Manuscript prepared for Nonlin. Processes Geophys.
with version 2014/09/16 7.15 Copernicus papers of the LaTeX class copernicus.cls.
Date: 28 April 2016

# Multiscale statistical analysis of coronal solar activity

Diana Gamborino[1], Diego del-Castillo-Negrete[2], and Julio J. Martinell[1]

[1]Instituto de Ciencias Nucleares, UNAM,, A. Postal 70-543, México D.F., Mexico
[2]Oak Ridge National Laboratory,, Oak Ridge, TN, 37831, USA

*Correspondence to:* J.J. Martinell (martinel@nucleares.unam.mx)

**Abstract.** Multi-filter images from the solar corona are used to obtain temperature maps which are analyzed using techniques based on proper orthogonal decomposition (POD) in order to extract dynamical and structural information at various scales. Exploring active regions before and after a solar flare and comparing them with quiet regions we show that the multiscale behavior presents distinct statistical properties for each case that can be used to characterize the level of activity in a region. Information about the nature of heat transport is also to be extracted from the analysis.

## 1 Introduction

The increasing number of space telescopes and space probes that provide information about phenomena occurring in the space is yielding enormous amount of data that need to be analyzed to get information about the physical processes taking place. In particular, images of the Sun obtained from the Solar Dynamics Observatory (SDO) instruments and the new Interface Region Imaging Spectrograph (IRIS) mission have a remarkable high resolution that allow studies of the Sun with great detail, not available before. The analysis of the images usually involves some processing if one has to extract information not readily obtained from the raw images. Some techniques used for image analysis and feature recognition in the Sun have been described by Aschwanden (2010). A useful tool that gives information about the physical conditions in the emitting region is the emission measure (EM) which is obtained from images in several wavelengths; this in turn provides the local temperature of the emitting region (e.g., Aschwanden and Boerner , 2011). When the EM is computed for all the pixels in a given region of the Sun, the temperature maps can be obtained. There are some computational tools developed in SolarSoftWare (SSW) that take images from different instruments at the available filters to produce EM and temperature maps (e.g. for Yohkoh, SOHO, TRACE, SDO).

The images can be processed for pattern recognition using techniques such as wavelet analysis that sort the structures in terms of the scaling properties of their size distribution (Farge and Schneider , 2015). Another method, which is also frequently used in image compression, is the singular value decomposition (SVD) of the pixel matrix. This method **is equivalent to the proper orthogonal decomposition (POD) that** has been applied in many contexts including: *solar physics* (Vecchio et al. , 2005, 2008) *image processing* (Rosenfeld and Kak , 1982); and *turbulence models* (Holmes et al. , 1996). Applications to plasma physics relevant to the present work include compression of Magnetohydrodynamics data del-Castillo-Negrete et al. (2007); detection of coherent structures (Futatani et al. , 2009), and multi-scale analysis of plasma turbulence (Futatani et al. , 2011; Hatch et al. , 2011). In this work we apply POD techniques to a time sequence of images representing temperature maps of the solar corona. The **implementation of the method that incorporates time and space variations, was referred to as Topos-Chronos by Aubry el al. (1991) and has been used in several studies to perform spatio-temporal analyses of turbulence (Benkadda et al. , 1994; Futatani et al. , 2011), and of some solar features (Carbone et al. , 2002; Mininni et al. , 2002, 2004; Lawrence et al. , 2005; Podladchikova, et al. , 2002; Vecchio et al. , 2009) under different names. Here we apply it to study the space-time evolution of different solar coronal regions.** The analysis of solar flare activity is of particular interest as it is the more energetic phenomena in the Sun linked with structural changes that may be detected with this method. To this end we perform an analysis of temperature maps corresponding to a region of solar flare activity, and compare the results with the analysis done in the *same* region *before* the solar flare and in a "quiet" region where no flares were detected during the time of observation. Our POD multi-scale study is supplemented with a statistical analysis of the probability distribution function (PDF) of temperature fluctuations. In particular, in the search for statistical precursors of solar flare activity, we show that the small scale temperature fluctuations in pre-flare states exhibit broader variability (compared to the flare and quiet sun states) and the corresponding PDFs exhibit non-Gaussian, stretched-exponential tails characteristic of intermittency. We also present a study heat transport during solar flare activity based on a simplified analysis that allows to infer the transport coefficients (advection velocity and diffusivity profiles) from the data.

The rest of the paper is organized as follows. In section 2 the fundamentals of POD methods are presented and the features of the Topos-Chronos method as the optimal representation of a truncated matrix are explained. Section 3 describes how the data for temperature maps are obtained for a solar flare event and for other solar regions. The results of applying the POD methods to the data are described in section 4 highlighting the statistical features of the multi-scale analysis. This leads to the identification of intermittency in an active region previous to a flare. As a further application, section 4.2 presents the evaluation of heat transport coefficients based on the full temperature maps and on POD. Finally, in section 5 we give the conclusions of the analysis and an appraisal of the POD methods applied to solar physics.

## 2 Description of POD methods used in the work.

In this section we review the basic ideas of the POD method and its application in separating spatial and temporal information as used in the present work.

### 2.1 SVD

The Singular Value Decomposition (SVD) is, generally speaking, a mathematical method based on matrix algebra that allows to construct a basis in which the data is optimally represented. It is a powerful tool because it helps extract dominant features and coherent structures that might be hidden in the data by identifying and sorting the dimensions along which the data exhibits greater variation. **In our case, SVD is used to decompose spatio-temporal data into a finite series of separable modes of time and space, which are orthonormal. The modes give the best representation of the relevant time and space scales of the data.**

The SVD of the matrix $A \in \mathbb{R}^{m \times n}$ is a factorization of the form,

$$\mathbf{A} = \mathbf{U}\Sigma\mathbf{V}^T \tag{1}$$

where $U \in \mathbb{R}^{m \times m}$ and $V \in \mathbb{R}^{n \times n}$ are orthogonal matrices (i.e. $U^T U = V V^T = I$) and $\Sigma \in \mathbb{R}^{m \times n}$ is a rectangular diagonal matrix with positive diagonal entries. The $N$ diagonal entries of $\Sigma$ are usually denoted by $\sigma_i$ for $i = 1, \cdots, N$, where $N = \min\{m, n\}$ and $\sigma_i$ are called singular values of A. The singular values are the square roots of the non-zero eigenvalues of both $AA^T$ and $A^T A$, and they satisfy the property $\sigma_1 \geq \sigma_2 \geq \cdots \geq \sigma_N$.(Golub and van Loan , 1996; Martin and Porter , 2012)

Another useful mathematical expression of SVD is through the tensor product. The SVD of a matrix can be seen as an ordered and weighted sum of rank-1 separable matrices. By this we mean that the matrix $A$ can be written as the external tensor product of two vectors: $u \otimes v$ or by its components: $u_j v_k$, i.e.,

$$A = \sum_i^r \sigma_i u_i \otimes v_i^T \tag{2}$$

where $r = \text{rank}(A)$. Here $u_i$ and $v_i$ are the i-th columns of U and V respectively. Equation (2) is convenient when one wants to approximate $A$ using a matrix of lower rank. In particular, according to the Eckart-Young Theorem (Eckart and Young , 1936), given eq. 1, the truncated matrix of rank $k < r = \text{rank}(A)$

$$A^{(k)} = \sum_{i=1}^k \sigma_i u_i \otimes v_i^T. \tag{3}$$

is the optimal approximation of $A$ in the sense that

$$\| A - A^{(k)} \|^2 = \min_{\text{rank}(B)=k} \| A - B \|^2$$

90 over all matrices $B$, where $\parallel A \parallel = \sqrt{\sum_{i,j} A_{ij}^2}$ is the Frobenius norm.

In our analysis the matrix $A(x, y)$ is the scalar field representing the temperature **or the energy content** at each point $x$, $y$ at a given time. **Thus, we have a time sequence of** maps that are represented by the 2D spatial rectangular grid $(x_i, y_j)$ where $i = 1, \ldots, N_x$ and $j = 1, \ldots, N_y$. The maximum number of elements in the SVD expansion (eq. (3)) is, therefore, $k = \min\{N_x, N_y\}$. These

95 maps change as a function of time.

## 2.2 Topos and Chronos

Topos and Chronos is **a name given to the the separation of time and space variations in the data in a way that the POD method can be applied** (Aubry el al. , 1991; Benkadda et al. , 1994). To illustrate this technique, consider a spatio-temporal scalar field $A(\boldsymbol{r}, t)$ representing the temperature

100 T, where $\boldsymbol{r} = (x, y)$. We introduce a spatial grid $(x_i, y_j)$ with $i = 1, \ldots, n_x$ and $j = 1, \ldots, n_y$ and discretize the time interval $t \in (t_{in}, t_f)$, as $t_k$ with $k = 1, \ldots, n_t$, $t_1 = t_{in}$ and $t_{n_t} = t_f$. We represent the data as a 3D array $A_{ijk} = A(x_i, y_j, t_k)$. Since the SVD analysis applies to 2D matrices, the first step is to represent the 3D array as a 2D matrix. This can be achieved by "unfolding" the  bf rows of the 2D matrix of the spatial domain of the data into a single row 1D vector, i. e. $(x_i, y_j) \rightarrow \boldsymbol{r}_i$

105 with $i = 1, \ldots, n_x \times n_y$, and representing the spatio-temporal data as the $(n_x n_y) \times n_t$ matrix $A_{ij} = A(r_i, t_j)$, with $i = 1, \ldots, n_x \times n_y$ and $j = 1, \ldots, n_t$.

The singular value decomposition of $A(r_i, t_j)$ is given by the tensor product expression,

$$A_{ij} = \sum_{k=1}^{N^*} \sigma_k u_k(r_i) v_k(t_j), \tag{4}$$

where $N^* = min[(n_x n_y, n_t)]$. In this sense, the POD represents the data as a superposition of sepa-

110 rable space time modes.The vectors $u_k$ and $v_k$ satisfy the orthonormality condition given by,

$$\sum_{i=1}^{n_x n_y} u_k(r_i) u_l(r_i) = \sum_{j=1}^{n_t} v_k(t_j) v_l(t_j) = \delta_{kl}. \tag{5}$$

The $n_x n_y$-dimensional modes $u_k$ are the "topos" modes, because they contain the spatial information of the data set, and the $n_t$-dimensional modes $v_k$ are the "chronos" since they contain the temporal information.

115 From eq. (3), we define the rank-$r$ optimal truncation of the data set $A^r$ as,

$$A_{ij}^{(r)} = \sum_{k=1}^{r} \sigma_k u_k(r_i) v_k(t_j). \tag{6}$$

where $1 \leq r \leq N^*$. The matrix $A^{(r)}$ is the best representation of the data due to the fact that, by construction, the POD expansion minimizes the approximation error, $\parallel A - A^r \parallel^2$.

## 3   Temperature data maps.

**The data we analyze with the methods of previous section are obtained from the observations of the Atmospheric Imaging Assembly (AIA) instrument (Boerner et al. , 2012; Lemen et al. , 2012) of the SDO, using the six filters that record the coronal emission. We are interested in the information related to the thermal energy distribution in the corona, which in general is difficult to obtain accurately due to the temperature sensitivity of the emission and radiation**

**transfer processes across the corona. Most of the methods used to obtain temperatures in the solar atmosphere rely on the isothermal assumption. This is a coarse approximation of the solar corona that might not be fully justified in the case of flaring activity. However, following Aschwanden et al.  (2013) we adopt this approximation in order to give a simple intuitive physical interpretation of the data. One alternative to avoid potential unintended consequences of**

**the isothermal assumption would be to check the width of the temperature distribution, and discard all pixels for which the assumption does not hold. However implementing this filter is outside the scope of the present manuscript that aims to explore the data directly using the proposed tools. Moreover, since we are interested in the thermal energy distribution in the corona and not on the absolute values of the temperature, we only need the relative values of**

**the thermal content and for this we take the "pixel-average" temperature obtained from the method developed by Aschwanden et al.  (2013) that was implemented in a SolarSoft routine. From the combination of the six filters dual maps for the emission measure (EM) and temperature are obtained. Alternate methods for inferring the temperature have also been developed by Guennou et al.  (2012) and Dudok de Wit et al.  (2013). Despite the approximations made,**

**our analysis has an intrinsic value independent of the level of justification regarding any given specific physical interpretation of the data themselves. Our goal is to extract valuable spatio-temporal dependencies given the data that is currently available using the simplest physical assumptions.**

The first step is to select appropriate events that contain the phenomenology of interest. In our case,

we focus on the propagation of a heat front associated with an impulsive release of energy, such as a solar flare. **What we call a heat front is simply an emitting thermal structure that moves across the solar disk, but we are not interested in the actual identification of it with known waves in the solar atmosphere. There could be various possibilities for the propagating front such as EIT waves (Gallagher and Long , 2011), coronal waves related to chromospheric Moreton**

**waves (Narukage et al. , 2004) or others, which are known to perturb some structures like in the wave-filament interactions (Liu el al. , 2013). It is, however, not relevant to our studies to know which type of perturbation is seen.** We look for events satisfying the following criteria: (1) Flares of medium intensity (classification C according to Geostationary Operational Environmental Satellite (GOES) ) for which no saturation of the image occurs. (2) Significant **contrast of the flare**

**related structures emission, in order to have a clear identification. This selects active-region**

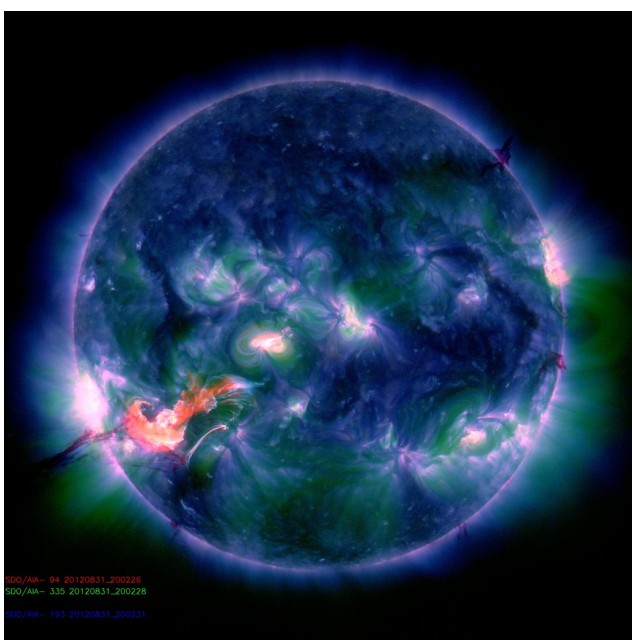

**Figure 1.** AIA Composite of EUV emissions (94, 193, 335 Å) [Ref.: http://sdo.gsfc.nasa.gov].

**events which are also compared to the same active region before the event and to a quiet sun region at the same period.**

The chosen event occurred on 31/8/2012 at around 20:00 UT. There were seven active regions on the visible solar disk. An image at a fixed time is shown in Fig. 1, formed from a compound of extreme ultraviolet (EUV) emissions (94,193, 335 Å) from the solar corona. Active region 11562 was the originator of the solar flare of GOES-class C8.4.

The AIA instrument consists of four detectors with a resolution of $4096 \times 4096$ pixels, where the pixel size corresponds to a space scale of $\approx 0.6''$ ($\approx 420$ km). It also contains ten different wavelengths channels, three of them in white light and UV, and seven in EUV, whereof six wavelengths (131, 171, 193, 211, 335, 94 Å) are centered on strong iron lines (Fe VIII, IX, XII, XIV, XVI, XVIII), covering the coronal range from $T \approx 0.6$ MK to $\gtrsim 16$MK. AIA records a full set of near-simultaneous images in each temperature filter with a fixed cadence of 12 seconds. More detailed instrument descriptions can be found in Lemen et al. (2012) and Boerner et al. (2012).

**Each wavelength is produced by a temperature distribution that peaks at a characteristic temperature ($T_\lambda$), so the intensity of the line for an emitting region is related to the relative contribution of this temperature.** Thus, the images of the different intensities, $F_\lambda(x, y)$, can be used to obtain maps of the coronal temperature. These intensities represent the measured photon flux, which is determined by **the emission process, the physical conditions in the region traversed by the photons** and the filter response. **The physical conditions affecting the radiation are not**

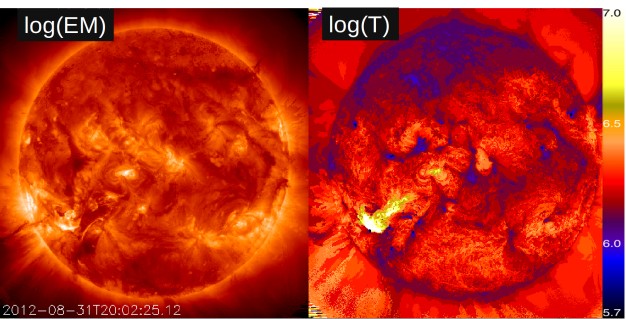

**Figure 2.** Emission Measure map (left) and Temperature map (right).

**known in general and are** usually represented by the so-called *differential emission measure* (DEM) which is used to obtain the temperature distribution of the plasma averaged along the line of sight.

The DEM distribution $[dEM/dT]$ is typically given by $dEM(T) = n^2 dh/dT$ [cm$^{-5}$ K$^{-1}$], where $n(h(T))$ is the electron density at height $h$ and with temperature $T$. The DEM distribution DEM$= d[EM(T,x,y,z)]/dT$ can be reconstructed from the six filter fluxes $[F_\lambda(x,y)]$, and in (Aschwanden et al. , 2013) developed a method to compute the peak emission measure (EM) and the peak temperature $T_p$ for each pixel based on a simple representation of the DEM by a Gaussian function of temperature, for each filter. Then, model fluxes are computed from the expression

$$F_\lambda(x,y) = \int DEM(T,x,y)R_\lambda dT$$

where $R_\lambda(T)$ are the filter response functions at a given wavelength, which are fitted to the observed flux $F_\lambda^{obs}(x,y)$. This determines the best-fit values for EM and $T_p$ thus obtaining the temperature maps. Here we use the Automated Emission Measure and Temperature map routines developed by Aschwanden et al. (2013) that implement the method of multi-Gaussian functions based on six-filter data, following a forward-fitting technique. The result of applying this post-processing methodology on the data in Fig. 1 is shown in Fig. 2. **Notice that the colors show values of $\log(T)$. Due to the uncertainties mentioned above, these maps are not true temperature maps but they provide information about the energy content distribution in the 2D space, and we can call them temperature-like maps**. The "temperature´´ map covers a temperature interval in the range $\log(T) = 5.7 - 7.0$ shown in the vertical bar. It highlights the hot regions while the EM map is more sensitive to the plasma density, since it is proportional to the square of the electron density (independent of temperature). Temperature maps were computed for a time span of 16.4 minutes with a cadence of 12 seconds, as provided by the AIA data. In total, 82 maps were generated. **For the relatively short duration, solar rotation is unimportant.**

## 4  Results of the analysis

The **maps** generated in the previous section combining the emission at six wavelengths can be further processed to extract structures and dynamical features **related to the thermal energy content**. The time evolution of the data space array, $A(x, y, t)$, is analyzed with the POD method described in Section 2. We first focus on the heat front that propagates from left to right in the region close to the flare site. From these maps we can extract information concerning: (1) the nature of the heat transport and (2) the physical conditions in the region. In particular, we seek to obtain information from the analysis about the energy distribution in the different spatial scales and how this compares with other regions with no flares. Fig. 3 presents the region where a heat front is moving at selected times in the full time interval; the vertical lines indicate the position of the leading-front at the initial and final times. We limit the attention to this region which is a square with dimensions $N_x \times N_y$ with $N_x = N_y = 32$ pixels (in these maps one pixel is $2.4" \times 2.4"$). **The length traveled by the thermal front set the size of all the regions analyzed, included the pre-flare and quiet sun cases, so they can be compared directly.** The POD method can be applied in a systematic way to separate time and space features and study them with a multi-scale analysis, or to determine the underlying features of heat transport.

The same POD analysis performed on the temperature maps in Fig. 3 is also applied to two other temperature maps corresponding to (a) exactly the same physical region but about one hour before the appearance of the flare (to which we refer as *pre-flare*) and (b) a different region in the quiet sun for the same time interval as the first map (referred to as *quiet-sun* or QS). These other analyses are made in order to compare the properties of the three cases and be able to determine the features associated with flare activity in the solar corona.

### 4.1  Multiscale statistical analysis using Topos-Chronos decomposition

As mentioned in subsection 2.2, the POD represents the data as a superposition of separable space-time modes. Figures 4, 5 and 6 show the result of applying this process to the three cases mentioned: after Solar flare, Pre-flare and Quiet Sun, respectively. The figures include behavior for small ranks, $k = 1, 2, 3, 4$ and large ranks, $k = 20, 30, 40, 46$. The 2D plots in the first and third columns present the spatial distribution, topos modes $u_k(r_i)$, rescaled by the singular values $\sigma^k$. The second and fourth columns show the chronos POD modes $v^k(t_j)$. One can see from Fig. 4 for the flare time, that, **as expected,** low-$k$ chronos modes exhibit a low frequency variation while large-$k$ modes exhibit high frequency burst activity. A similar behavior is found for the topos modes in which large scale structures are seen for low $k$ while a granulated structure with small scales is found for high $k$. **Although this is a natural consequence of the Topos-Chronos decomposition, it is of interest to determine the degree of correlation between space scales and time scales as the rank $k$ changes**.

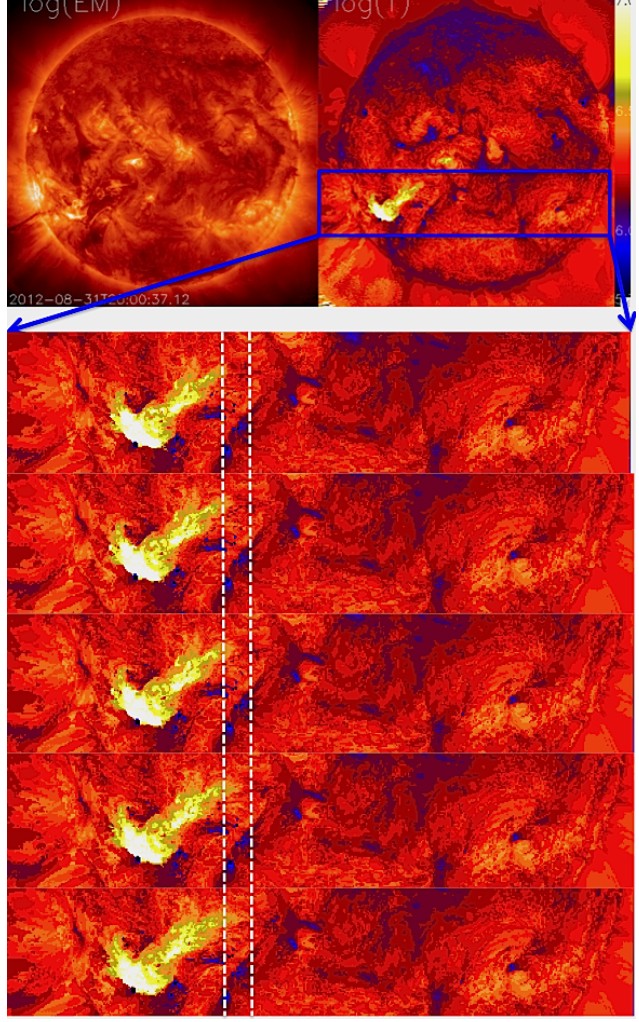

**Figure 3.** Snapshots at different times of the region where the heat front is moving.

This correlation can be studied using Fourier analysis to extract the characteristic spatiotemporal scales of each POD mode. For the Topos modes, we first transform the one-dimensional vector $r_i$ back to space coordinates $x_l$ and $y_m$ and compute the two-dimensional Fourier transform, $\hat{u}_k(\kappa_{x_l}, \kappa_{y_m})$. The characteristic length scale of the topos rank-$k$ mode is then defined as the mean length scale of the corresponding Fourier spectrum, $\lambda(k) = 1/\langle \kappa_k \rangle$, where $\langle \kappa_k \rangle$ is defined as:

$$\langle \kappa_k \rangle = \frac{\Sigma_{i,j} |\hat{u}_k(\kappa_{xi}, \kappa_{yj})|^2 (\kappa_{xi}^2 + \kappa_{yj}^2)^{1/2}}{\Sigma_{i,j} |\hat{u}_k(\kappa_{xi}, \kappa_{yj})|^2} \tag{7}$$

Using the same procedure, we associate a characteristic time scale $\tau$ to each chronos $v_k(t_m)$ mode using $\tau(k) = 1/\langle f_k \rangle$ with:

$$\langle f_k \rangle = \frac{\Sigma_m |\hat{v}_k(f_m)|^2 f_m}{\Sigma_m |\hat{v}_k(f_m)|^2} \tag{8}$$

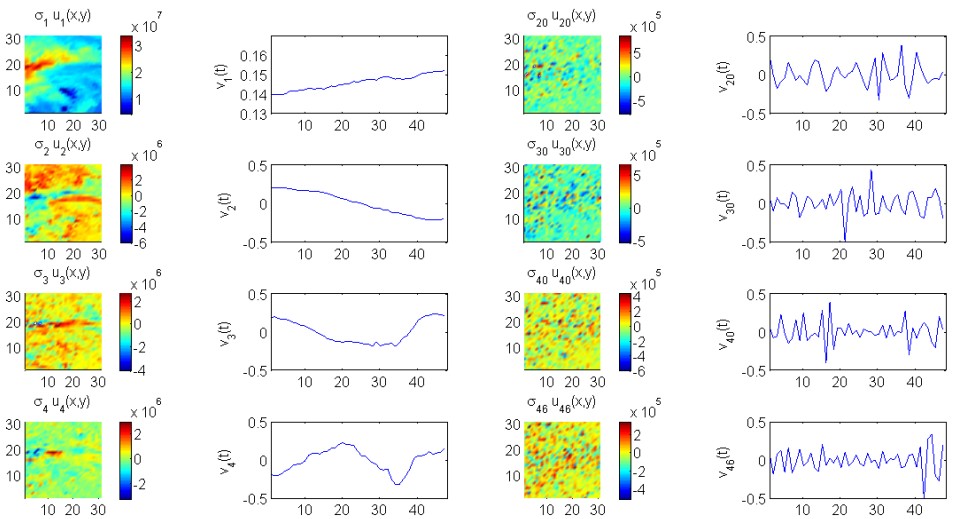

**Figure 4.** Topos-Chronos decomposition of solar flare activity. Odd columns: spatial modes $u_k(r_i)$; even columns: temporal modes $v_k(t_j)$, for k=1,2,3,4,20,30,40,46. One unit in the time axis equals 12 s.

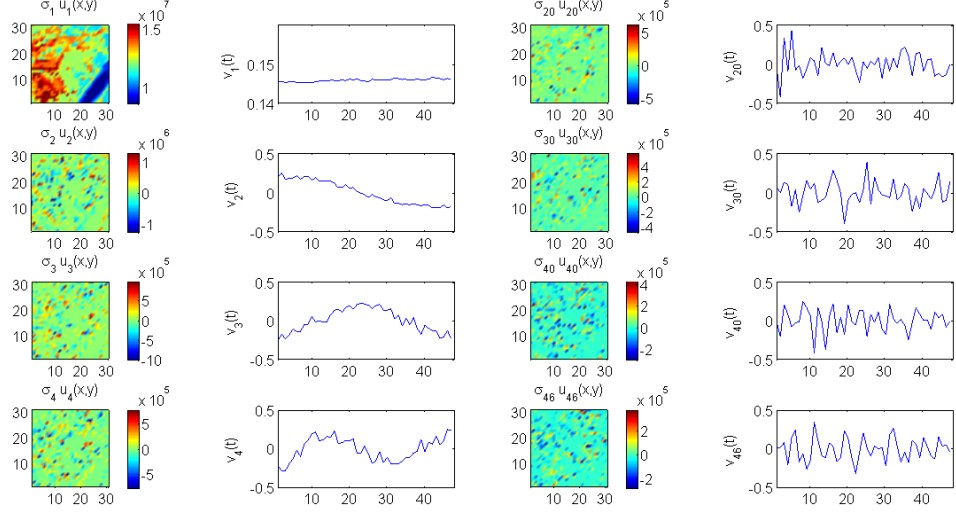

**Figure 5.** Topos-Chronos decomposition of pre-flare solar activity. Odd columns: spatial modes $u_k(r_i)$; even columns: temporal modes $v_k(t_j)$, for k=1,2,3,4,20,30,40,46. One unit in the time axis equals 12 s.

where $\hat{v}_k(f_m)$ is the Fourier transform in time of $v_k(t_m)$. Figure 7 shows the characteristic length scale $\lambda(k)$ versus the characteristic time scale $\tau(k)$ for all ranks. The tendency for increasing time scale with increasing space scales is apparent, although there is some dispersion for small space-time scales. The scaling can be fitted by a power law of the type $\lambda \sim \tau^\gamma$ and in this case $\gamma = 0.11$. For


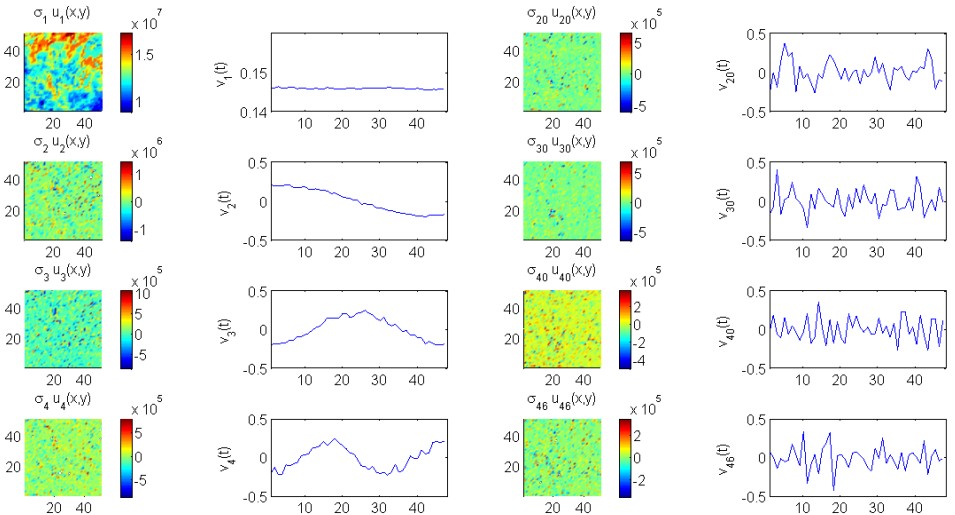

**Figure 6.** Topos Chronos decomposition of quiet sun region. Odd columns: spatial modes $u_k(r_i)$; even columns: temporal modes $v_k(t_j)$, for k=1,2,3,4,20,30,40,46. One unit in the time axis equals 12 s.

a diffusive-like process $\gamma = 1/2$, thus this results suggests that a sub-diffusive heat transfer may be taking place.

The topos-chronos plots for the pre-flare and quiet-sun cases in Figs. 5 and 6 do not show a clear correlation of the time and space scales. One can notice that only for rank 1 there are dominant large spatial scales while for higher ranks there are only small scales present. In the chronos diagrams the dominant small frequencies seen at low $k$ have higher frequency oscillations superimposed. In order to determine if there is any correlation between spatio-temporal scales we make the same Fourier

analysis as for the case with flare. The resulting length and time scales for all ranks are shown in Figure 8 for the two regions, pre-flare and QS. It is clear that for these cases no correlation is found and thus, it is not possible to attribute any cascading-like process, as in the region affected by the flare.

Further information can be obtained from the reconstruction error in the POD representation,

defined as the difference $\delta T_{t_i}^{(r)} = T_{t_i} - T_{t_i}^{(r)}$ where $T$ is the original temperature map at a given time $t_i$, and $T_{t_i}^{(r)}$ is the reconstructed map up to rank-$r$ at the same time $t_i$. Since a measure of the total energy content in the data is given by $E = \sum_{ij} A_{ij}^2 = \sum_{k=1}^{N^*} \sigma_k^2$, the energy contained in the reconstruction to rank-$r$ is

$$E(r) = \sum_{k=1}^{r} \sigma_k^2$$

with $\sigma_k^2$ giving the partial energy contribution of the $k$th-mode. Thus, the reconstruction error $|\delta T_{t_i}^{(r)}|^2$ contains the remaining energy in the leftover ranks $E - E(r)$. In particular, for high $r$, $\delta T_{t_i}^{(r)}$ is as-

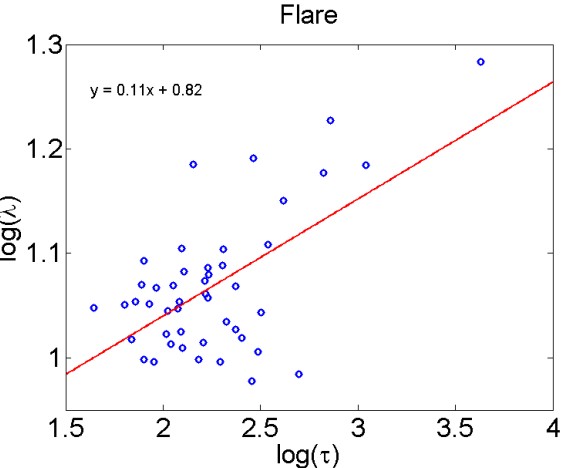

**Figure 7.** Length scale vs time scale for all POD ranks; Solar flare.

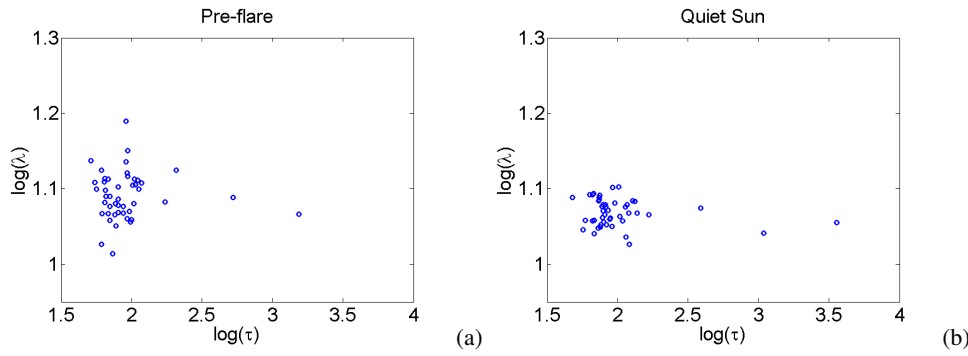

**Figure 8.** Length scale vs time scale for all POD ranks: (a) Pre-flare and (b) Quiet Sun.

sociated with the energy in the small scales. Given that the singular values, satisfying $\sigma_k \leq \sigma_{k+1}$, decay very fast in the first few modes, as seen in the POD spectra of Figure 9(a) for the three cases analyzed, most of the energy is contained in the low-$r$ reconstructions. To see this, Figure 9(b) shows the energy contribution percentage, $E(r)/E$, as a function of the reconstruction rank-$r$ for the three cases under study. It is clear that the first mode already contains about two thirds of the energy and successive modes contribute less as the rank rises. The case with the flare has the energy distributed over more modes, reflecting the effect of the large scale flows produced by the flare.

To quantify the degree of the temperature intermittency, we construct the probability distribution function (PDF) by dividing the range of values of $T - T_{t_i}^{(r)}$ in small bins and counting the number of grid points of the map falling in each bin. The resulting histogram represents the PDF of $T - T_{t_i}^{(r)}$ and this can be done for different percentages of the energy contribution, i.e. taking the corresponding rank $r$ from Fig. 9(b). We chose to use two energy fractions to compare them, namely a residual energy of 10% and 25% (i.e. $1 - E(r)/E$). For each case, there is one PDF for each time $t_i$ having its own height and width, but we expect them to have similar statistical features. Thus, as it is customary,

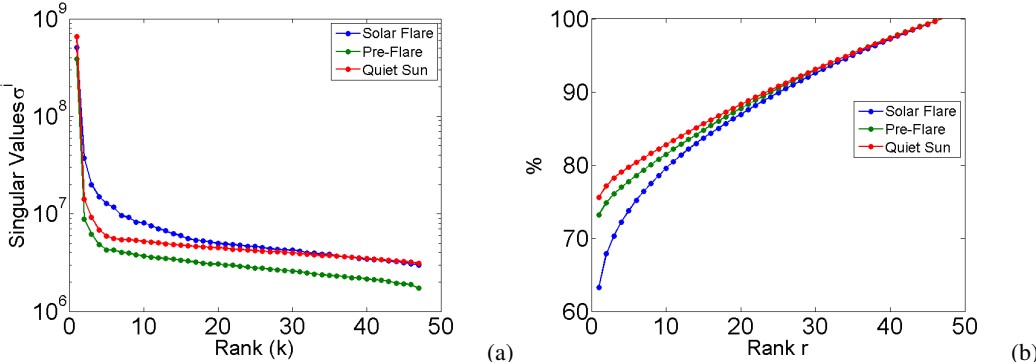

**Figure 9.** (a) Singular values spectra $\sigma^k$ as a function of $k$. (b) Energy contribution percentage of the rank-$r$ reconstruction.

we rescale the PDF as $P \rightarrow \sigma P$ and $x \rightarrow x/\sigma$ where $\sigma$ is the standard deviation. Since $\sigma$ changes with the different time frames, it is more convenient to use the average $\langle\sigma\rangle_t$ over all times in order to have a single normalizing parameter. Thus, for every time the PDFs were rescaled with the average standard deviation as $P \rightarrow \langle\sigma\rangle_t P$ and $\delta T_{t_i}^r \rightarrow \delta T_{t_i}^r / \langle\sigma\rangle_t$. It is found that the resulting rescaled PDFs

are all quite similar having a small overall dispersion. Then we took the average over the time sequence of PDFs (47 frames) in order to have a single PDF that represents the statistical properties of the reconstruction error maps for each energy content. The results for the two energy percentages are shown in figure 10. It is observed that the PDFs of the pre-flare region are significantly broader than the other two, indicating the presence of very large temperature fluctuations.

In order to determine if the PDFs have non standard features like long tails, it is useful to fit some known function to the data. Normal statistics produce a Gaussian PDF, so we can fit a stretched exponential $P(x) \sim \exp(-\beta|x|^\mu)$ to the average PDFs in order to asses how far they are from a Gaussian ($\mu = 2$). In Figure 11 we show the best fits to each of the six cases analyzed. For each PDF we present a fit of the central part and another fit of the tails (with smaller $\mu$). Clear differences are

observed among the Flare, Pre-flare and QS cases. However, for the different energy contents, the results do not show great variability. The interesting result is that pre-flare PDFs, in addition to being the broadest of the three (see Fig. 10), have the lowest values of $\mu$ and so they are the ones that depart the most from a Gaussian; the long tails indicate there is intermittency, i.e. a relatively large number of intense temperature fluctuations. On the other hand, the PDFs for the Flare have larger $\mu$ and thus

are closer to a Gaussian; large fluctuation events are then less frequent. Our results agree with the study in Futatani et al. (2009) that observed lower levels of intermittency in the presence of sources. The work in Futatani et al. (2009) followed the transport of impurities in a plasma with decaying turbulence (without sources to sustain it) finding evidence of intermittency, but it disappears when there are sources that maintain the turbulence. In our case, the flare plays the role of a source that

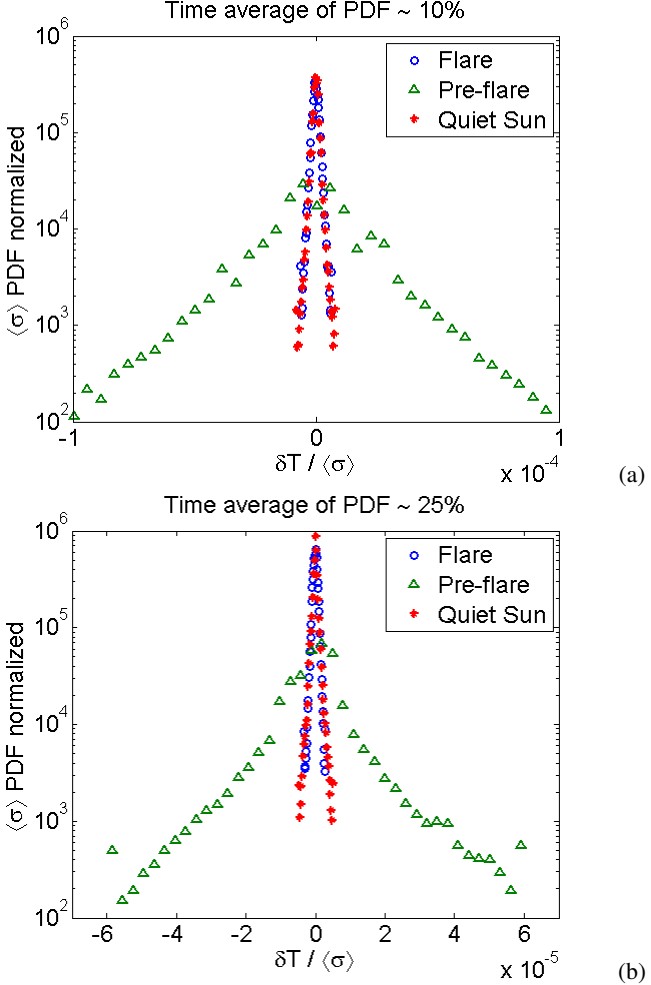

**Figure 10.** PDF averaged over time frames for energy content in the small scales of (a) 10 % (more reconstruction modes) and (b) 25 % (less reconstruction modes)

suppresses intermittency. As expected for a region with no solar activity, in the quiet sun case, the PDF does not exhibit long tails, i.e., the probability of large temperature fluctuations is small.

### 4.2 Heat Flux Estimate

In this section we focus on the region affected by the flare and calculate the thermal flux entering there, driven by the released flare energy. This is made using the time evolution of the original temperature maps and comparing the results with those obtained with the Topos-Chronos method. Next, the heat flux is used to explore the properties of heat transport in the corona.

Since the thermal front produced by the flare is a coherent structure it is expected that the main mechanism bringing energy into the region of interest is advective. Using this premise, we use the computed heat flux to estimate the velocity of the incoming plasma flow, as a function of the coor-

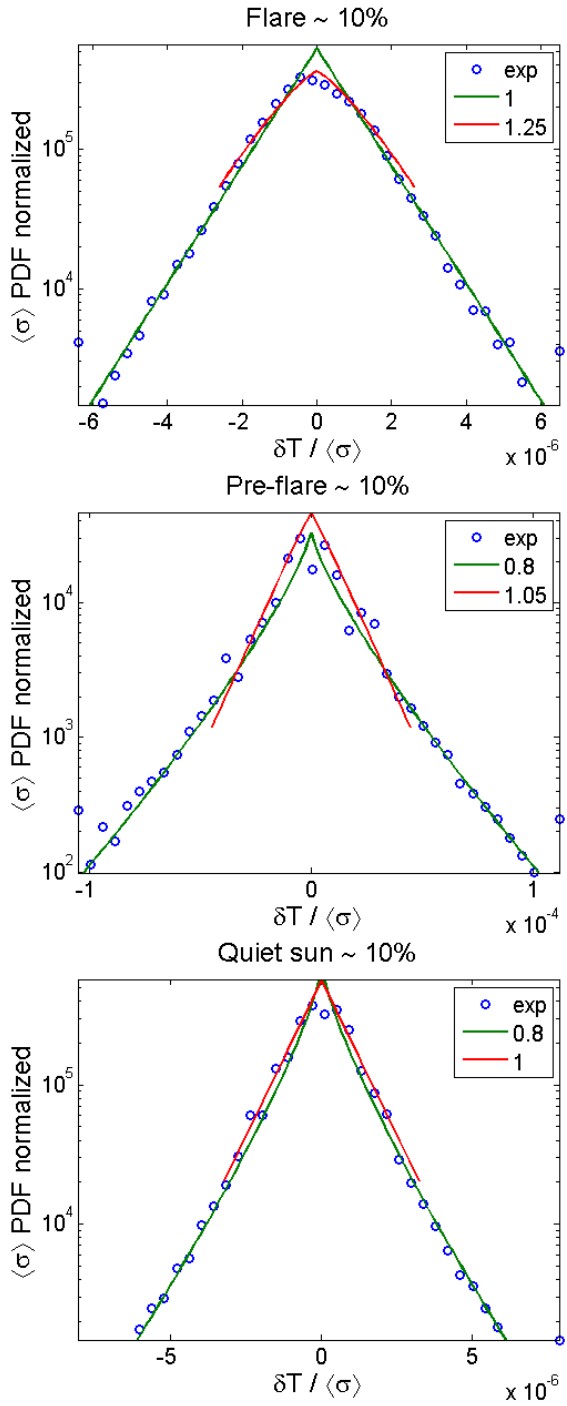

**Figure 11.** Stretched exponential fits of the temperature fluctuations PDFs when energy content in small scales is 10 %; $\mu$ values are given in the legends. (a) Flare [$\mu = 1, \beta = 9.75 \times 10^5$ and $\mu = 1.25, \beta = 1.82 \times 10^7$ ], (b) Pre-flare [$\mu = 0.8, \beta = 9 \times 10^3$ and $\mu = 1.05, \beta = 1.3 \times 10^5$ ] and (c) Quiet Sun [$\mu = 0.8, \beta = 9.1 \times 10^4$ and $\mu = 1, \beta = 10^5$ ].

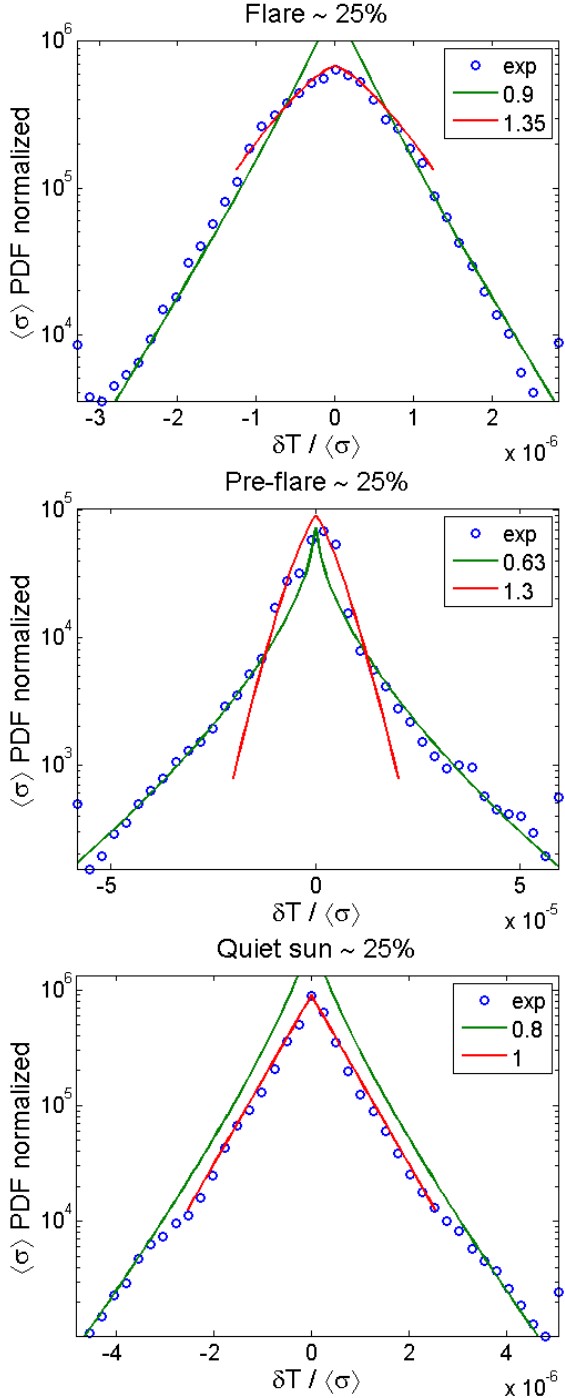

**Figure 12.** Stretched exponential fits of the temperature fluctuations PDFs when energy content in small scales is 25 %; $\mu$ values are given in the legends. (a) Flare [ $\mu = 0.9, \beta = 6.3 \times 10^5$ and $\mu = 1.35, \beta = 9 \times 10^4$ ], (b) Pre-flare [ $\mu = 0.63, \beta = 2.8 \times 10^3$ and $\mu = 1.3, \beta = 1.5 \times 10^8$ ] and (c) Quiet Sun [ $\mu = 0.8, \beta = 1.5 \times 10^5$ and $\mu = 1, \beta = 1.7 \times 10^6$ ].

dinate $x$ along the flow (the velocity profile $V(x)$). However, there might be a diffusive component

playing part in the process. To investigate this, we compare the heat flux profile with the temperature

and temperature gradient profiles to look for convection or diffusion relations. Assuming that both

processes coexist, we estimate velocity profiles taking different levels of diffusion. We also explore

the scenario of diffusion being the main transport process by computing the profiles of the diffusivity

$D = D(x)$, but we find that it is not viable since $D$ becomes negative.

For the analysis of the heat flux we make the following assumptions: (1) there are no sources of

energy inside the region of the temperature maps since it was chosen to be slightly away from the

flare site; (2) the main direction of motion is along the $x$-axis ($y$ displacements are negligible and

no information of the dynamics in the $z$-axis is known: we have an integrated view along $z$); (3)

the internal energy of the plasma is $u = \frac{3}{2} n k_B T$ and no mechanical work is done by the plasma; (4)

we use a constant electron density with the value for the lower solar Corona: $n = 10^{15}$ m$^{-3}$ in the

region of interest.

**Total Heat Flux using Original T-maps:** The starting point is the heat transport equation in the

absence of sources:

$$\frac{3}{2} n k_B \frac{\partial T}{\partial t} = -\nabla \cdot \boldsymbol{q} \tag{9}$$

Since the thermal front moves almost exclusively in the $x$ direction, we assume that the $y$ variations

are not very relevant and therefore we take the average of eq. (9) over $y$. Moreover, we are interested

in the energy transport over the whole time interval, $\tau$, and therefore we take the time average of

eq.(9)

$$\frac{3}{2} n k_B \left\langle \overline{\frac{\partial T}{\partial t}} \right\rangle_y = -\langle \overline{\nabla \cdot \boldsymbol{q}} \rangle_y \tag{10}$$

where $\langle f \rangle_y = \frac{1}{L} \int_0^L f \, dy$ is the average value in the $y$-direction and $\overline{f} = \frac{1}{\tau} \int_0^\tau f \, dt$ is the time average.
Integrating eq.(10) along the $x$-coordinate and using that there is no flux through the upper and lower
egdes, i.e. $\langle \partial q_y / \partial y \rangle_y = 0$, we get:

$$\frac{3}{2} n k_B \int_x^L \left\langle \overline{\frac{\partial T}{\partial t}} \right\rangle_y (x') dx' \quad = \quad \int_x^L \frac{\partial}{\partial x'} \langle \overline{q_x} \rangle_y dx'$$

$$= \quad \langle \overline{q_x} \rangle_y (L) - \langle \overline{q_x} \rangle_y (x)$$

Assuming that the first term on the right hand side is zero, meaning there is no heat leaving the

region at $x = L$, the average heat flux profile is

$$\langle \overline{q_x} \rangle_y (x) = \frac{3}{2} n k_B \int_x^L \langle \Delta_\tau T \rangle_y dx'. \tag{11}$$

where $\Delta_\tau \xi \equiv (\xi(t = \tau) - \xi(t = 0))/\tau$.

In Figure 13 we show the heat flux profile computed from the original temperature maps using

(11), **for the averaged 1D model**. The heat flux can have an advective and a diffusive component

$\mathbf{Q} = W\mathbf{v} - D\nabla W$ (we do not consider here radiative transport) with $W = nk_BT$ the thermal energy density and $D$ the heat diffusivity. The relative importance of these contributions can be assessed by computing the profiles for the temperature and the temperature gradient and comparing them with the heat flux profile. The temperature profile is taken by averaging the original temperature maps in $t$ and $y$ and similarly for the temperature gradient. These are also shown in Fig. 13 which indicates that the $T(x)$ profile is similar to the $Q(x)$ profile suggesting that the advective component should be the dominant one. In contrast, the negative temperature gradient has no clear correlation with $Q(x)$ which seems to indicate that diffusion is not the main drive for the heat flux **along** $x$ **in this particular event**. From these observations we can compute the advection velocity as

$$v = \frac{Q + Dnk_B\nabla T}{nk_BT}$$

In Figure 14 the velocity profile computed in this way is shown for the case with no diffusion ($D = 0$) and when a constant diffusivity is assumed to be present ($D_0 = (5$ and $15) \times 10^{10}m^2s^{-1}$). Diffusivities larger than $1.5 \times 10^{11}m^2s^{-1}$ will give sign changes in $v$ which is unlikely to occur in a heat front, **so this value sets an upper limit to** $D$. **This limit is consistent with the values found with other methods which are in the range** $10^2 - 10^4 km^2/s$ **(Aschwanden , 2012). We** 340     could also explore the possibility of detecting diffusive transport by computing the diffusivity, $D = (v - Q/nk_BT)T/\nabla T$. **When this is done it is found that** $D$ **has negative values at several points along** $x$ **which indicates that diffusive transport in the** $x$ **direction is subdominant**. There can only be a small contribution superimposed on the advective transport. **However, diffusive transport can be noticeable in the** $y$ **direction where an important advection is not present. This can be** 345     **studied with a similar analysis by averaging accrosss the** $x$ **direction. Notice that the averaging procedure is a simplification that would produce only approximate results. The equivalent of Eq. 11 is**

$$\langle\overline{q_y}\rangle_x(y) = \frac{3}{2}nk_B\left[\langle\Delta_\tau T\rangle_{xy}\,y - \int_0^y \langle\Delta_\tau T\rangle_x\,dy'.\right] \tag{12}$$

**where it was used again that** $\overline{q_y}$ **vanishes at** $y = 0, L$ **and** $\langle.\rangle_{xy}$ **is the double average over** $x$ **and** 350     $y$. **Eq. 11 was used to express** $\overline{q_x}$ **at** $x = 0$. **Using these estimates, Figure 15 shows the averaged heat flux profile across** $y$, **together with averaged temperature and temperature gradient profiles. In this case, no correlation can be observed between** $\overline{q_y}$ **and the temperature and thus the advection is not a dominant process as it was for the flux along** $x$. **The temperature gradient has large fluctuations due to the averages in** $x$ **and time of a widely varying temperature dis-** 355     **tribution. However, there is some consistency with the heat flux, like the vanishing values of both at the edges and a rough correlation near the central part of the region. Therefore, we can assert that diffusion is probably the dominant process in the** $y$ **direction. A diffusion coefficient cannot be estimated because of the weak correlation resulting from our approximation. The order of magnitude obtained for the region where there is a correlation turns out to be too**

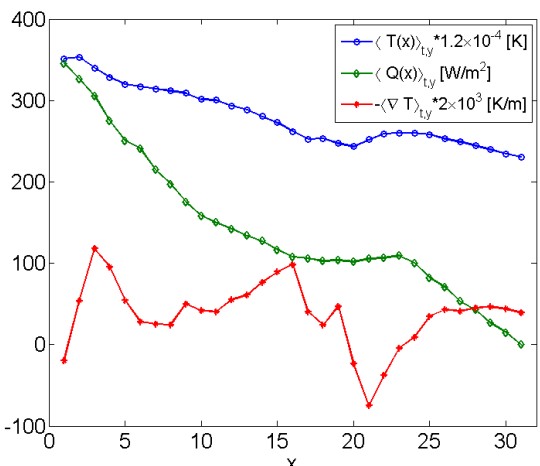

**Figure 13.** Averaged (in $t$ and $y$) profiles calculated from the original full temperature maps, for temperature (Blue); heat flux (Green); and negative temperature gradient (Red).

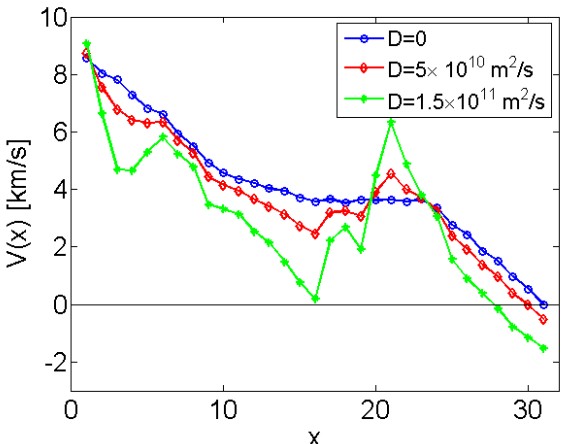

**Figure 14.** Velocity profiles assuming different levels of constant diffusivity.

**large, of order $10^{11} m^2/s$. But since advection is not a candidate, diffusive transport is then an acceptable model for propagation perpendicular to the heat front direction.**

**Heat Flux using Topos-Chronos:**

In addition to the decomposition of the temperature fluctuations in optimal modes, the Topos-
Chronos method can be used to do a multi-scale analysis of transport. The separation of space and time allows the extraction of prominent spatial structures persistent over the time span, ordered by rank. Similarly, temporal representation gives information about the time evolution of the structures

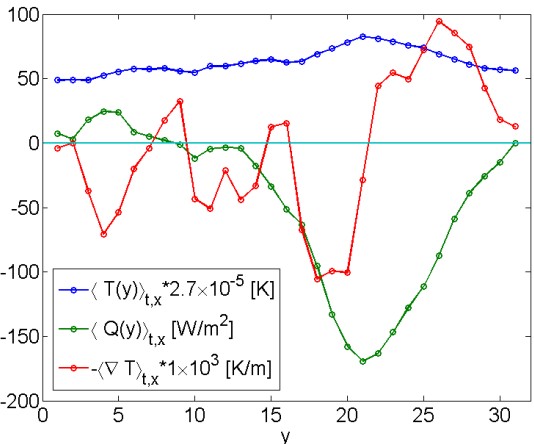

**Figure 15.** Averaged (in $t$ and $x$) profiles calculated from the original full temperature maps, for temperature (Blue); heat flux (Green); and negative temperature gradient (Red)..

at the corresponding rank. This separation is amenable to compute the dominant spatial temperature gradients, that drive the heat fluxes, and the time variation of the thermal energy, independently.

Substituting the representation for the temperature maps in eq. 4 into eq.(11) we obtain the Topos-Chronos decomposition of the heat flux profile **along x**:

$$\sum_{k=1}^{N^*} \langle \overline{q_x} \rangle_y(x) = \frac{3}{2} n k_B \int_x^L \left\langle \Delta_\tau \sum_{k=1}^{N^*} \sigma^k u^k(r_i') v^k(t_j) \sigma^k \right\rangle_y dx'$$

The time derivative as well as the time average affect only the temporal modes; the integral and the average in $y$ affect only the spatial modes. When the space components $u^k(r_i)$ are transformed back

to a 2D array using the coordinate transformation $r_i \to (x_l, y_m)$, we obtain the matrix $\hat{u}^k(x_l, y_m)$. Each mode of the heat flux is thus given by

$$\langle \overline{q_x} \rangle_y^k(x) = \frac{3}{2} n k_B \sigma^k \Delta_\tau v^k \int_x^L \left\langle \hat{u}^k(x', y) \right\rangle_y dx'. \tag{13}$$

The heat flux profiles for the first four modes are shown in Figure 16. As expected, the dominant mode $k = 1$ has a profile similar to that of the raw temperature maps in Fig. 13. For the other ranks

the flux is smaller but, interestingly, for $k = 2$, $Q$ is negative, indicating there is a second order return heat flow.

  A similar analysis can be made for the $k = 1$ mode. The result is presented in Figure 17 where, as before, a correlation between $Q(x)$ and $T(x)$ is observed, but not between $Q(x)$ and the temperature gradient. Thus, it is again concluded that advection is the dominant mechanism and the advection ve-

locity can be computed in the same way. The resulting velocity profile is shown in Figure 18 together with the velocity obtained from the original maps for comparison. The Topos-Chronos velocity is larger in most of the region because it lacks the contributions from higher modes, especially the

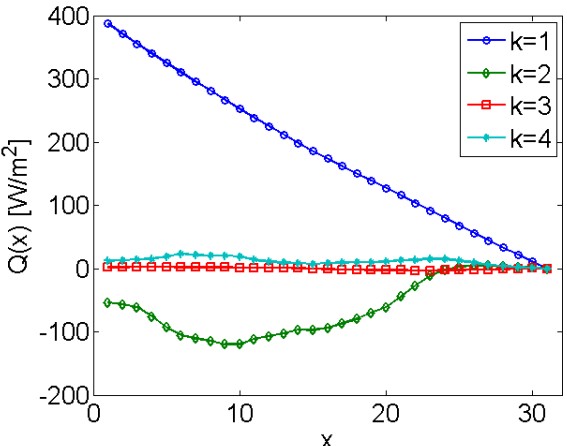

**Figure 16.** Heat flux profiles as a function of $x$ for four values of the rank $k$.

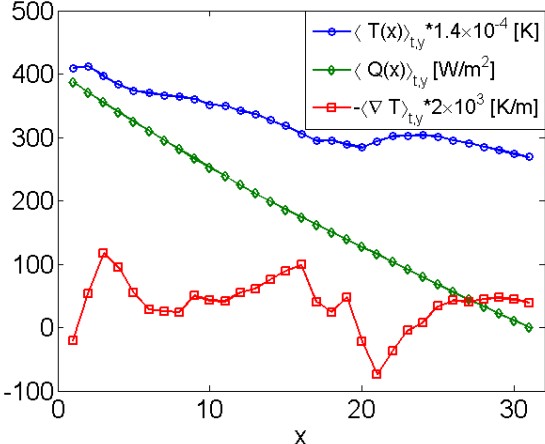

**Figure 17.** Averaged (in $t$ and $y$) profiles calculated using Topos-Chronos for rank $k=1$, for temperature (Blue); heat flux (Green); and negative temperature gradient (Red).

negative $k = 2$ rank. **But notably, this shows that for the first mode the heat front has a uniform deceleration all along the path; higher modes are responsible for the small variations. While the $k = 1$ mode has not the complete information, it reveals the presence of some underlying uniform transport porcess.** Higher modes could also provide information about the small scale diffusive transport. However, a quantitative analysis to estimate the diffusivity cannot be done since, as before, it turns out to give negative values at some points. This is because taking the average over the $y$ direction is not a good procedure for this purpose since the contribution of small scales is washed out and this is important for higher modes. The same statement holds for transport along $y$ averaged over $x$.

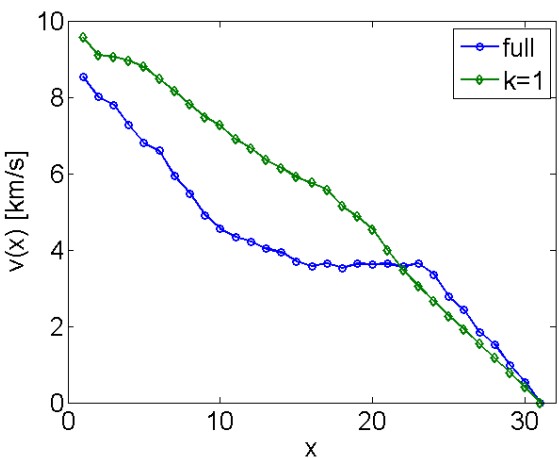

**Figure 18.** Velocity profiles assuming different levels of constant diffusivity.

## 5    Discussion and Conclusions

**Using EUV images from six filters of the SDO/AIA and simple physical assumptions (mainly isothermality within a pixel) we have constructed maps representing the energy content 2D dis-**
**tribution that we use as approximate temperature maps for the Solar Corona.** The temperature maps for a solar event near an AR in the Solar Corona were analyzed using POD methods and were compared with similar analyses of other maps: one of the same region but before the flare (pre-flare) and another of a quiet region during the same time period (quiet sun). The POD method separates time and space information (Topos-Chronos) thus allowing to determine the dominant space-time scales. The high-rank modes in the decomposition correspond to smaller spatial scales and in some degree with small time-scales. An interesting finding is that there is a rough correlation between Fourier time and spatial scales when the flare has occurred but not for pre-flare or QS states. This may indicate that flare-driven large scale heat flows tend to transfer their energy to smaller scales.

The Topos-Chronos method was also used to study the statistical properties of the temperature fluctuations. In particular, we reconstructed the temperature maps up to a certain rank, associated with a given energy content. The reconstruction error was thus associated with the small-scale temperature fluctuations. The probability distribution functions (PDFs) of these small-scale fluctuations were obtained for all times and then averaged in time for each of the three regions analyzed. One of the main results of the present paper is the observation that the PDF for the pre-flare state is significantly broader and has longer tails than the PDFs of the flare and quiet sun cases. The pre-flare activity seems to produce more high-amplitude temperature fluctuations, characteristic of intermittency, which might herald the occurrence of the flare. **It is interesting to note that this result may be related to the findings of Abramenko et al. (2003) who also showed that there is evidence of intermittency in the magnetic field of an active region previous to the occurrence of a flare.**

**They argue that this indicates that there is a turbulent phase before the flare, which would be in agreement with the intermittency in the temperature fluctuations found here.**

A multi-scale analysis of the heat flux was also performed for the region associated with the flare. The thermal flux profiles along the main ($x$) direction of the flow were computed using the original temperature maps and compared with the temperature variation along $x$, allowing to obtain the

advection velocity profile. Diffusive transport is found to be **sub-dominant and cannot be evaluated. A similar analysis was performed for transport along the direction perpendicular to the heat front propagation and in that case diffusion shows as a considerable contribution to transport.** The same computations were applied for the Topos-Chronos decomposition finding also a prevalence of advection with a $v(x)$ profile for the lowest mode that agrees with the one for the

original maps **and suggesting the presence of an underlying uniform transport process**. Higher modes are not so relevant for the advective flux but can provide information about small scale diffusion. **We point out that indications about a diffusive-like transport associated with a solar flare have been found by Aschwanden (2012) who actually found that the transport is sub-diffusive. This agrees with our result of Fig.7 which shows that the correlation of time and space scales**

**corresponds with a sub-diffusive process.**

*Acknowledgements.* We thank Dr. Alejandro Lara for his assistance with the use of SolarSoft and IDL. We acknowledge the use of a SolarSoft pachage developed by M. Aschwanden to obtain temperature maps. This work was partially supported by DGAPA-UNAM IN109115 and CONACyT 152905 Projects. D dCN acknowledges support from the Oak Ridge National Laboratory, managed by UT-Battelle, LLC, for the U.S. Department of

Energy under contract DE-AC05-00OR22725.

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
