# Peer review of "Multiscale statistical analysis of coronal solar activity"

_Nonlinear Processes in Geophysics, 2015_

## Referee Comment (RC1) · Anonymous Referee #1 · 28 Jan 2016

This study addresses the spatio-temporal dynamics of the solar corona from a interesting point of view, by decomposing coronal temperature maps over a given time interval into small sets of separable modes, similar to what had been pioneered in the 1990s in neutral fluid turbulence by N. Aubry, R. Lima, P. Holmes, G. Berkooz, and more.

However, this study looks much like a replication of [Futatani et al., Phys. Plasmas 16, 042506 (2009)] to a solar dataset, without properly taking into account the strong assumptions behind this dataset. In addition, the method is over-interpreted, thus leading to incorrect conclusions. Because there are several major issues here, I cannot recommend this manuscript for publication.

Here are some general comments

Your study heavily relies on the physically appealing concept of coronal temperature.

[Figure]

However, the temperature maps as provided by the method of Aschwanden et al. involve several strong assumptions. In particular, these temperature maps are assuming an isothermal plasma, which is a coarse approximation of what the true corona is; it certainly does not hold during flaring activity, and it is likely too to be incorrect too during transients, such as your heat fronts. In addition, flares cause artefacts such as pixel bleeding. Another good reason to be very careful when interpreting synoptic EUV images.

These limitations (which are mentioned in Aschwanden's article) should be considered very seriously before any physical interpretation can be given to these temperature maps. A first and obvious starting point would be to check the width of the temperature distribution, and discard all pixels for which the isothermality assumption does not hold.

Following this, I'm deeply concerned that the whole interpretation in Sections 3 to 5 remains purely speculative as long as these spurious effects are properly addressed, which may be quite challenging. Personally, I would refrain from using temperature maps at all, except for qualitative analysis, or for structures that are known to be approximately isothermal.

What is then the best observable ? Note that most studies consider log(T) rather than T because it is more convenient, and also because the distribution of T is assumed to be log-normal. Several of the properties of the SVD are optimal for datasets that have a normal distribution. For that reason, I would seriously consider working with log(T) rather than with T – assuming of course that the temperature can be used at all.

I also have major concerns regarding your interpretation of the SVD results. In line 178, you say that "there is some correlation between small scales and high frequencies". This is merely a consequence of the properties of the SVD, and has nothing to do with the physics. Whenever you diagonalise a covariance matrix (what the SVD does, in some way) whose values decay monotonically as you move away from the diagonal, then the eigenmodes (your topos and chronos) will be like Fourier modes,

whose number of nodes will increase with the rank k. So, small wavenumbers will automatically be associated with small frequencies. Try to generate a surrogate dataset that has the same second order properties, and you'll get exactly the same results. For that same reason, there is no evidence whatsoever for a cascading (line 193). Incidentally, because the SVD exploits second order moments only, I would not recommend in problems in which phase coherence matters.

Later on (line 195) you introduce the scaling index gamma: this makes no sense because several of your topos and chronos don't have a clear characteristic size, or time scale. You may find values for <kappa> and <f>, but this does not prove that they make sense as they would, for example, for a wavelet basis. For example, for the rank 1 mode you capture the average background temperature, whose spatial or temporal scale is of no particular interest here, should definitely be excluded from your analysis. Several more details suggest that you're over-interpreting what the SVD is telling. I strongly recommend that you check your results carefully and test them, in particular by using surrogate data. This also applies to Figure 9, from which one cannot draw serious conclusions without knowing what the confidence intervals of the singular values and energy spectra are.

I would have many more comments, also regarding the analysis of the pdfs. Let me mention one more, in line 150: what is it you are calling a heat front ? Flares are intense events that generate various types of transients. So-called EIT waves have received considerable attention [Gallagher and Long, Space Science Reviews, 158 (2011), pp. 365–396] but they're not the only ones. Here, I strongly recommend that you put your analysis in context, and emphasize the novelty of your results in comparison to existing studies.

And now some more specific comments:

title: your title is misleading as you are not truly doing a multiscale analysis. The SVD does indeed separate different scales, but these are very loosely defined, and

are in no way comparable to what truly multiresolution techniques, such as the wavelet decomposition, would give.

sun –> Sun

several citations do not show up correctly, e.g. line 24

line 17: obtained –> inferred

line 26: the POD is strictly identical to the SVD, not more general

line 27: many more studies have used the SVD, or variants thereof to investigate the spatio-temporal dynamics of the Sun. I would be good to mention some of them, and not focus only on the work by Vecchio et al.

line 33: topos and chronos are not a method, but just the names given to the spatial, resp. temporal modes obtained by applying the SVD to a spatiotemporal dataset, see [N. Aubry et al., Journal of Statistical Physics, 64 (1991), pp. 683–739]. BTW, in that context, the SVD is called biorthogonal decomposition.

line 33 the specific method: all these methods (POD, biorthogonal decomposition, SVD, EOF, PCA ...) are identical; what distinguishes them to some degree is the type of data they are applied to, or the preprocessing, but even that is not always true. So there is no point mentioning them as if they were different. Otherwise people keep on reinventing the wheel.

line 35: why is flaring activity interesting ? section 2.1 this part is clearly written, but quite mathematical, and devoid of a connection with the physics. It would help to say the your spatio-temporal wakefield gets decomposed into a finite series of separable modes of time and space, which, in addition, are orthonormal, etc.

line 88: mention at least the original work by Aubry, Lima, et al, who coined the words topos and chronos.

line 94: unfolding –> folding end of 2.1: again, what is the physical motivation behind

working with Aˆ{(r)} ?

line 119: why "requires significant emission of radiation" ?

line 120: what is the time span of your data set, and did you correct for solar rotation ? The latter point is *very* important because the properties of your SVD modes change if your spatial frame is moving.

line 131: For each wavelength there is a corresponding temperature: this is incorrect. Each wavelength is associated with a temperature distribution.

line 134: Do you mean that the filter response is associated with a DEM ?

line 135: note that there are alternate methods for inferring the temperature, such as [Guennou et al., Astrophysical Journal Supplement Series 203 (2012)], and [Dudok de Wit et al., Solar Physics, 283 (2012), pp. 31–47].

line 158: why that particular size of 32x32 ? Why not larger or smaller ? What is limiting the number of time steps ? Notice that since Nx*Nx » Nt, in your covariance matrix your ensemble average is done along the spatial dimension, and not along the more usual temporal one. This impacts your results, and should be addressed.

Fig. 4: what are there oblique stripes in all of your pictures, as if the plasma was moving sideways ? Since your spatial region is a square, it would make more sense to force its aspect ratio to 1.

line 175: why plot the absolute value ? and how should it better reveal periodicities ?

line 193: see general comments. There is no cascading whatsoever here.

line 199: the mode with rank 1 is just the average background. Usually, when analyzing a spatio-temporal wavefield that is quasi-stationary (as is the case here, as T stays around 10ˆ6 C), that first mode should be discarded since it doesn't tell us much about the dynamics. What matters is the variation on top of it.

line 208: T should also be indexed by t_i

line 210: for the reason explained just above, since you're interested by the dynamics only, you should start by subtracting the time-average from each pixel. This will affect the distribution of the singular values. The wording "energy" will then make much more sense as it truly describes the variance of the modes.

Fig. 10: please use symbols that can be read on B/W printouts.

Sections 4 and 5 : to be considered once the problem with these temperature maps has been addressed.

---

## Referee Comment (RC2) · Anonymous Referee #2 · 3 Feb 2016

SUMMARY AND GENERAL EVALUATION

The authors present an interesting spatio-temporal analysis of 4-dim solar data, for a flaring event, the preflare phase, and a Quiet Sun region. They use a Singular Value Decomposition method and investigate the multi-scale behavior of the physical parameter of the temperature T, and attempt to extract information on the heat flow dT/dx in a flare. The description of the method is clear, and the data analysis steps are described in sufficient detail. In the second half of the paper the authors try to extract information on the heat flux before and during a flare, where they arrive at the result that diffusive transport is not relevant in solar flares, which is at odds with other studies. If the authors can reconcile the interpretation of this controversial result in terms of the limitations of the used method, the paper may possibly be suitable for publication.
[Figure]

DETAILED COMMENTS

In the Discussion and Conclusions the authors arrive at several important and some controversial results:

(1) "... a raw correlation between Fourier time scale and spatial scales during the flare, but not for pre-flare or quiet Sun. This may indicate that the flare-driven heat flows tend to decay into smaller scales, in a cascade-like process." - Perhaps a reference and discussion can be made whether this is relevant to the scenario of "inverse MHD turbulent cascade" (e.g., Abramenko et al. 2003, ARep 47, 151, "Pre-Flare Changes in the Turbulence Regime for the Photospheric Magnetic Field in a Solar Active Region"; Antonucci et al. 1996, ApJ 456, 833 " Interpretation of the observed plasma turbulent velocities as a result of magnetic reconnection in solar flares"; LaRosa,T.N. and Moore,R.L. 1993, ApJ 418, 912-918, "A mechanism for bulk energization in the impulsive phase of solar flares: MHD turbulent cascade".

(2) "... The pre-flare activity seems to produce larger low-amplitude fluctuations, characteristic of intermittency, which might herald the occurrence of the flare." — Discuss in this in the context of the paper by Abramenko et al. (2003), for instance.

(3) "A multi-scale analysis of the heat flux was also performed for the region associated with the flare. The thermal flux profiles along the main direction (x) of the flow were computed original temperature maps and compared with the temperature variation along x, allowing to obtain the advection velocity profile. Diffusive transport is found to be not relevant." — This is contrary to other observational findings that the envelope of the flare area propagates like diffusive transport (Aschwanden 2012, ApJ 757, 94, "The spatio-temporal evolution of solar flares observed with AIA/SDO: Fractal diffusion, sub-diffusion, or logistic growth". The author's result may be biased because they use a 1-dim transport model (in x-direction), with no transport in transverse direction. Furthermore, they assume that there is no energy source inside the region of temperature maps (line 270). It would be more realistic to assume that the energy

transport is isotropic in all directions, because the high-resolution images from AIA show that a flare entails many reconnection sites with magnetic field lines that go in all directions. Perhaps the authors can modify their heat flux model accordingly, or at least discuss the limitations of their model and what bias is expected in the diagnostics of the overall heat flux from a (point-like) flare source region.
* * *

---

## Author Comment (AC1) · 27 Apr 2016

We thank the referee for the comments and we attach a PDF file with our detailed answers. The document gives a sequence of the referee's original comments, followed by our answer and the corresponding modification of the text. We hope the answers and the changes can be convincing. The paper has been greatly improved as a result of this revision. The new version will be uploaded as soon as the system requests us to do so, according to the journal procedures.

Please also note the supplement to this comment:
http://www.nonlin-processes-geophys-discuss.net/npg-2015-83/npg-2015-83-AC1-supplement.pdf

[Figure]

**Supplement:**

We thank the referee for his (her) valuable comments and suggestions. Following is our response and the changes incorporated in the new version of the manuscript. To make the revision more transparent, we have grouped the referee's comments by topics.

**Opening paragraph:**

Referee's comment:

  "This study addresses the spatio-temporal dynamics of the solar corona from a interesting point of view by decomposing coronal temperature maps over a given time interval into small sets of separable modes, similar to what had been pioneered in the 1990s in neutral fluid turbulence by N. Aubry, R. Lima, P. Holmes, G. Berkooz, and more. However, this study looks much like a replication of [Futatani et al., Phys. Plasmas 16, 042506 (2009)] to a solar dataset, without properly taking into account the strong assumptions behind this dataset. In addition, the method is over-interpreted, thus leading to incorrect conclusions"

Authors' response:

We appreciate that the referee finds our approach to the spatio-temporal dynamics of the solar coronal interesting. However we do not agree on the statement that "…this study looks much like a replication of [Futatani, et al. (2009)] ….."  The fact that we use the similar data analysis tools does not make our work a replica. In particular, the problem studied in the paper under submission has its own interest since we use actual solar data whereas the problem addressed in [Futatani, et al.  (2009)] deals with numerical simulations. We do not agree either with his (her) remarks that the method has been over-interpreted and/or that our conclusions are incorrect. These issues, as well as several others, are addressed below. Having said this, we acknowledge that several parts of our original manuscript needed further explanations and improvements, and we sincerely thank the referee for bringing to our attention points that needed extra work.

**General Comments:**

**Use of temperature maps**

Referee's comment:

 "Your study heavily relies on the physically appealing concept of coronal temperature. However, the temperature maps as provided by the method of Aschwanden et al. involve several strong assumptions."

Authors' response:

We agree with the referee that the interpretation of the temperature maps provided by the method of Aschwanden involves assumptions that need to be clarified, and we thank him for explicitly pointing this out. Addressing these concerns has improved our manuscript.

However, we want to stress that our analysis of these maps has an intrinsic value

independent of the level of justification regarding any given specific physical interpretation of the data. That is, our main contribution is to provide a novel, unique analysis of data that we, as well as the scientific community, believe to be physically meaningful and related to the thermal properties of the solar corona. We assume (hope) that the referee is not implying that the data is meaningless and read his criticism as a valuable call to be more careful regarding the physical interpretation, something we completely agree.

Referee's comment:

"In particular, these temperature maps are assuming an isothermal plasma, which is a coarse approximation of what the true corona is; it certainly does not hold during flaring activity, and it is likely too to be incorrect too during transients, such as your heat fronts."

Authors' response:

We agree with the referee, and have modified the manuscript to explicitly mention the limitations of the interpretation of the data as temperature maps.

Changes incorporated:

Page 5 Line 120 added:

The data we analyze with the methods described in the previous section were obtained from the observations of the Atmospheric Imaging Assembly (AIA) instrument (Boerner et al. , 2012; Lemen et al. , 2012) of the SDO, using the six filters that record the coronal emission. We are interested in the information related to the thermal energy distribution in the corona, which in general is difficult to obtain accurately due to the temperature sensitivity of the emission and radiation transfer processes across the corona. Most of the methods used to obtain temperatures in the solar atmosphere rely on the isothermal assumption. This is a coarse approximation of the solar corona that might not be fully justified in the case of flaring activity. However, following Aschwanden et al. (2013) we adopt this approximation in order to give a simple intuitive physical interpretation of the data.

(4 lines below) Moreover, since we are interested in the thermal energy distribution in the corona and not on the absolute values of the temperature, we only need the relative values of the thermal content and for this we take the "pixel-average" temperature obtained from the method developed by Aschwanden et al. (2013) that was implemented in a SolarSoft routine. From the combination of the six filters dual maps for the emission measure (EM) and temperature are obtained.

(next line) Despite the approximations made, our analysis has an intrinsic value independent of the level of justification regarding any given specific physical interpretation of the data themselves. Our goal is to extract valuable spatio-temporal dependencies given the data that is currently available using the simplest physical assumptions.

Referee's comment:

 "In addition, flares cause artifacts such as pixel bleeding. Another good reason to be very careful when interpreting synoptic EUV images."

Authors' response:

We would like to point out that in the data considered in the present work the intensity of the signal is not high enough for the pixel bleeding to be a concern. We are not observing the flare at its peak intensity.

Referee's comment:

"These limitations (which are mentioned in Aschwanden's article) should be considered very seriously before any physical interpretation can be given to these temperature maps. A first and obvious starting point would be to check the width of the temperature distribution, and discard all pixels for which the isothermality assumption does not hold.

Authors' response:

Checking the width of the temperature distribution to discard data for which the isothermal assumption does not hold is a valuable suggestion to post-process the data before analyzing it with the proposed tools. However, this task is outside the scope of the present paper that aims to explore the data directly using the proposed tools using the simplest physical assumptions. Having said this, we have added this valuable suggestion in the manuscript.

Changes incorporated:

Page 5 Line 129 added:

One alternative to avoid potential unintended consequences of the isothermal assumption would be to check the width of the temperature distribution, and discard all pixels for which the assumption does not hold. However implementing this filter is outside the scope of the present manuscript that aims to explore the data directly using the proposed tools.

Referee's comment:

Following this, I'm deeply concerned that the whole interpretation in Sections 3 to 5 remains purely speculative as long as these spurious effects are properly addressed, which may be quite challenging. I would refrain from using temperature maps at all, except for qualitative analysis, or for structures that are known to be approximately isothermal."

Authors' response:

We believe that in our responses to the previous points we have made clear that the data we are using are meaningful; they are based on standard approximations made in the literature. Therefore the analyses made in Sections 3-4 are not speculative but reflect the presence of actual physical processes. Section 5 contains the conclusions and in response to the referee's multiple comments we have, once again, reiterated the role of the approximations involved.

Changes incorporated:

Page 21 Line 392 added:

Using EUV images from six filters of the SDO/AIA and simple physical assumptions (mainly isothermality within a pixel) we have constructed maps representing the energy content 2D distribution that we use as approximate temperature maps for the Solar Corona.

Referee's comment:

 "What is then the best observable? Note that most studies consider log(T) rather than T because it is more convenient, and also because the distribution of T is assumed to be log-normal. Several of the properties of the SVD are optimal for datasets that have a normal distribution. For that reason, I would seriously consider working with log(T) rather than with T – assuming of course that the temperature can be used at all."

Authors' response:

It is important to keep in mind that T and log(T) contain exactly the same physical information, and in this sense they are the same observable. Thus, deciding which one is "better" has not an objective answer. In those cases when T is assumed to be log-normal, we agree with the referee that working with log(T) might be more efficient. However, it is important to keep in mind that we are not assuming anything about the statistics of T. In fact, one of the key contributions of this work is to actually compute the PDF of T directly from the data, and to do this accurately we have found that working directly with T has advantages.  Having clarified this, we were intrigued by the referee's suggestion and have actually redone the multi-scale decomposition in Figs. 4, 5 and 6 and have observed that they are fundamentally the same for T and log(T) (see figures below as compared to Figs. 4, 5, 6 of the paper) which again confirms the fact that they contain the same information.

**Data analysis**

Referee's comment:

line 119: why "requires significant emission of radiation" ?

Authors' response:

We do not understand the Referee's question. Line 119 says "significant emission variation" it does not say "significant emission radiation".

line 120: what is the time span of your data set, and did you correct for solar rotation ? The latter point is *very* important because the properties of your SVD modes change if your spatial frame is moving.

Authors' response:

The time span of the event analyzed is 80 x 12 sec = 16 min. This is a relatively short time compared to the rotation period of the sun (~27 days). Therefore, there is no need to correct for solar rotation effects.

Changes incorporated:

Page 7 Line 189 added:

For the relatively short duration, solar rotation is unimportant.

Referee's comment:

line 131: For each wavelength there is a corresponding temperature: this is incorrect. Each wavelength is associated with a temperature distribution.

Authors' response:

We agree with the referee and thank him (her) for pointing this out. We have modified the manuscript accordingly.

Changes incorporated:

Page 6 Line 169 added:

Each wavelength is produced by a temperature distribution that peaks at a characteristic temperature, $T_\lambda$, so the intensity of the line for an emitting region is related to the relative contribution of this temperature.

Referee's comment:

line 134: Do you mean that the filter response is associated with a DEM ?

Authors' response:

No, this is not what was written in the manuscript. However, to avoid potential misunderstanding we have modified the sentence.

Changes incorporated:

Page 6 Line 174:

The physical conditions affecting the radiation are not known in general and are usually represented by the so-called differential emission measure (DEM) which is used to obtain the temperature distribution of the plasma averaged along the line of sight.

Referee's comment:

line 135: note that there are alternate methods for inferring the temperature, such as [Guennou et al., Astrophysical Journal Supplement Series 203 (2012)], and [Dudok de Wit et al., Solar Physics, 283 (2012), pp. 31–47].

Authors' response:

We thank the referee for bringing to our attention these references which we have added to the manuscript.

Changes incorporated:

Page 5 Line 138 added:

Alternate methods for inferring the temperature have also been developed by Guennou et al. (2012) and Dudok de Wit et al. (2013).

Referee's comment:

line 158: why that particular size of 32x32 ? Why not larger or smaller ? What is limiting the number of time steps ? Notice that since Nx*Nx » Nt, in your covariance matrix your ensemble average is done along the spatial dimension, and not along the more usual temporal one. This impacts your results, and should be addressed.

Authors' response:

The pixel number was determined by the size of the region where the moving front was present during the chosen time span which turned out to be 32X32. The number of time frames was limited by the selection criteria mentioned in section 3. We are not doing any ensemble average, our method follows the standard procedure of unfolding the space data in a vector to form the space-time matrix.

Indeed there is a larger number of space points Nx*Ny that time points Nt but there is no problem with this. The results we show do not involve any ensemble average.

Changes incorporated:

Page 8 Line 202 added:

The length traveled by the thermal front set the size of all the regions analyzed, included the pre-flare and quiet sun cases, so they can be compared directly.

Referee's comment:

Fig. 4: what are there oblique stripes in all of your pictures, as if the plasma was moving sideways ? Since your spatial region is a square, it would make more sense to force its aspect ratio to 1.

Authors' response:

Because these stripes show up in the topos diagrams they are not related to propagation events. Most likely they are due to a spatial correlation at the corresponding scale. The referee is right about the aspect ratio equals 1, we are changing it in the new version.

Changes incorporated:

Page 9 and 10, modified aspect ratio in Figures 4, 5 and 6.

Referee's comment:

line 175: why plot the absolute value ? and how should it better reveal periodicities ?

Authors' response:

We were looking for a correlation between periods of activity and the mode rank so the only information needed is the amplitude of $v^k(t\_j)$. However, since no correlation was found we decided that there is no need to mention that and therefore we have removed this comment from the new version.

**SVD analysis**

Referee's comment:

 "I also have major concerns regarding your interpretation of the SVD results. In line 178, you say that "there is some correlation between small scales and high frequencies". This is merely a consequence of the properties of the SVD, and has nothing to do with the physics. Whenever you diagonalise a covariance matrix (what the SVD does, in some way) whose values decay monotonically as you move away from the diagonal, then the eigenmodes (your topos and chronos) will be like Fourier modes whose number of nodes will increase with the rank k. So, small wavenumbers will automatically be associated with small frequencies.

Authors' response:

In the SVD the relevant scales for each rank come out as a result of the analysis. Therefore, although in a given problem wavenumbers and frequencies might increase with increasing rank they don't necessarily might do it uniformly or at the same rate.

Thus, any potential correlation between spatio-temporal scales cannot be a priori assumed and needs to be unveiled using the SVD. In fact, in the SVD analysis of the solar data, such correlation is not found in the pre-flare or quiet sun regions as seen in Fig. 8. An even more dramatic example is the analysis presented in [Futatani et. al. (2009)] (See Figs. 8, 11, 13 and 15 of that paper) where the correlation between the spatial and temporal scales for different ranks was shown to depend fundamentally on the intrinsic physical properties of the underlying turbulence. It is easy to construct examples of surrogate data sets that contradict the referee statement that this "is merely a consequence of the properties of the SVD, and has nothing to do with the physics". Consider for instance, a dataset generated from the function

$$f(x,y,t)=\sum_{j=1}^{N} A_j \cos(k_j(x+y))\cos(\omega_j t)$$ which represents the superposition of pure

modes with associated amplitudes $A_j$ that determine the energy content of each mode. Depending on how $A_j, k_j$ and $\omega_j$ are ordered the SVD would produce different correlations between space and time scales. To illustrate the point we constructed two datasets with just 10 modes (N=10) with amplitudes in decreasing order $A_j=(20,18,16,..,4,2,0.1)$ ; SVD will then order the modes according to $j$. When the wave numbers and frequencies are both monotonically decreasing $k_j, \omega_j=(10,...,1)$ one expects a direct correlation between time and space scales. When we apply the same processing used in the paper to this data, the plot obtained for the scales $\tau$ and $\lambda$ is shown in the next figure on the LHS. On the other hand, when the frequencies increase monotonically $\omega=(1,...10)$ keeping $k_i$ decreasing, there should be an inverse correlation, and this is indeed what is found as shown on the RHS of the figure. That shows that the relation between space and time scales is not merely a consequence of the SVD analysis but comes as a results of intrinsic properties of the data.

[Figure]

In the new version we limit to pointing out the different degrees of correlation for the three cases studied, but we do not mention any relationship with cascading, as indicated below.

Referee's comment:

Try to generate a surrogate dataset that has the same second order properties, and you'll get exactly the same results. For that same reason, there is no evidence whatsoever for a cascading (line 193). Incidentally, because the SVD exploits second order moments only, I would not recommend in problems in which phase coherence matters."

Referee's comment:

line 193: see general comments. There is no cascading whatsoever here.

Authors' response:

It is not clear what the referee means by second order properties or phase coherence. As explained below eq. 5 our use of SVD is based on finding the best approximation of data based on the Frobenius norm. This minimization process takes into account the whole spatial temporal dependence of the dataset. Having said this we agree that the interpretations of a cascading process are too preliminary at this point and needs further investigation. Accordingly we decided to remove this and address this interesting issue in a future publication.

Referee's comment:

 "Later on (line 195) you introduce the scaling index gamma: this makes no sense because several of your topos and chronos don't have a clear characteristic size, or time scale. You may find values for <kappa> and <f>, but this does not prove that they make sense as they would, for example, for a wavelet basis."

Authors' response:

We do not agree with the referee's statement that "topos and chronos do not have characteristic size" since the modes shown in the diagrams have an associated scale seen as the size of the small granulations in the topos part and a dominant frequency in the chronos part. Given this fact the Fourier modes do provide the characteristic scales.

Even though the correlation of spatio-temporal scales may not be interpreted as a cascade, the type of correlation does have some information that can be extracted when such a correlation exists. It conveys information about how the energy content in each rank is distributed in time and space scales. In other works it has been found to be diffusive-like  (gamma=1/2) or super-diffusive (gamma>1/2) (Futatani et al. 2009). When no correlations can be found (as in pre-flare and quiet sun cases) a fitting cannot be found and nothing can be concluded for those cases.

Referee's comment:

 "Several more details suggest that you're over-interpreting what the SVD is telling. I strongly recommend that you check your results carefully and test them, in particular by using surrogate data. This also applies to Figure 9, from which one cannot draw serious

conclusions without knowing what the confidence intervals of the singular values and energy spectra are."

Authors' response:

Figure 9 contains the information on how the method captured the amplitude of the different modes, and it is important to determine the relative weight of them. The analysis is based in a single space-time dataset folded into a matrix and there are no statistical variations that would require the assesment of a confidence interval.

Referee's comment:

"your title is misleading as you are not truly doing a multiscale analysis. The SVD does indeed separate different scales, but these are very loosely defined, and are in no way comparable to what truly multi-resolution techniques, such as the wavelet decomposition, would give."

Authors' response:

SVD and wavelets are both multi-scale analysis tools that depending on what you want to study. Each one has advantages and disadvantages; see for example Futatani et al. (2011).

The scale decomposition in the SVD is based on the energy content which is a well defined mathematical concept. One advantage of SVD is that it does not specify the mode structure a priori but it is determined as a result of the analysis. In that sense we feel the title is not misleading.

Referee's comment:

"the POD is strictly identical to the SVD, not more general"

Authors' response:

Not quite, in fact the SVD is a general mathematical technique used to factorize matrices that has been applied to many different problems beyond mode decomposition. In our context of data analysis, POD and SVD are indeed the same. To avoid controversies, we have removed any statement implying there is any difference between them.

Changes incorporated:

Page 2 Line 26 added:

This method (SVD) is equivalent to the proper orthogonal decomposition (POD) that has been applied in many contexts …

Referee's comment:

"many more studies have used the SVD, or variants thereof to investigate the spatiotemporal dynamics of the Sun. I would be good to mention some of them, and not focus only on the work by Vecchio et al."

Referee's comment:

"line 33: topos and chronos are not a method, but just the names given to the spatial, resp. temporal modes obtained by applying the SVD to a spatiotemporal dataset, see [N. Aubry et al., Journal of Statistical Physics, 64 (1991), pp. 683–739]. BTW, in that context, the SVD is called biorthogonal decomposition."

Referee's comment:

"line 33 the specific method: all these methods (POD, biorthogonal decomposition, SVD, EOF, PCA ...) are identical; what distinguishes them to some degree is the type of data they are applied to, or the preprocessing, but even that is not always true. So there is no point mentioning them as if they were different. Otherwise people keep on reinventing the wheel."

Referee's comments:

"line 88: mention at least the original work by Aubry, Lima, et al, who coined the words topos and chronos."

line 208: T should also be indexed by t_i

Fig. 10: please use symbols that can be read on B/W printouts.

Authors' response:

We agree with these comments, so we have modified the text accordingly in the right places.

Changes incorporated:

Page 2 Line 33 added:

The implementation of the method that incorporates time and space variations, was referred to as Topos-Chronos by Aubry el al. (1991) and has been used in several studies to perform spatio-temporal analyses of turbulence (Benkadda et al. , 1994; Futatani et al. , 2011), and of some solar features (Carbone et al. , 2002; Mininni et al. , 2002, 2004; Lawrence et al. , 2005; Podladchikova, et al. , 2002; Vecchio et al. , 2009) under different names. Here we apply it to study the space-time evolution of different solar coronal regions.

Referee's comment:

"line 94: unfolding –> folding end of 2.1: again, what is the physical motivation behind

working with Aˆ{(r)} ?”

Authors' response:

 "Unfolding" is correct because we mean converting an array of rows (matrix) to a single long row (vector) (one matrix row after the other). The motivation of working with Aˆ{(r)} has been included.

Referee's comment:

For example, for the rank 1 mode you capture the average background temperature, whose spatial or temporal scale is of no particular interest here, should definitely be excluded from your analysis.

Referee's comment:

line 199: the mode with rank 1 is just the average background. Usually, when analyzing a spatio-temporal wavefield that is quasi-stationary (as is the case here, as T stays around 10ˆ6 C), that first mode should be discarded since it doesn't tell us much about the dynamics. What matters is the variation on top of it.

Referee's comment:

line 210: for the reason explained just above, since you're interested by the dynamics only, you should start by subtracting the time-average from each pixel. This will affect the distribution of the singular values. The wording "energy" will then make much more sense as it truly describes the variance of the modes.

Authors' response:

The rank 1 mode has important information too, including time dependence; that is why we keep it. In fact the analysis on heat transport of Section 4.2 is mostly based on this mode. Subtracting the rank 1 would be misleading. The analysis mentioned by the referee would provide different information.

**Front and solar flares**

Referee's comment:

 "what is it you are calling a heat front ? Flares are intense events that generate various types of transients. So-called EIT waves have received considerable attention [Gallagher and Long, Space Science Reviews, 158 (2011), pp. 365–396] but they're not the only ones. Here, I strongly recommend that you put your analysis in context, and emphasize the novelty of your results in comparison to existing studies."

Author's response:

What we call a heat front is simply an emitting thermal structure that moves across the solar disk. The actual identification of the structure with any of the waves found in the solar atmosphere (EIT, Moreton waves, etc) is not important to us. However, a short discussion about the possible comparison with these waves has been included.

Changes incorporated:

Page 5 Line 146:

What we call a heat front is simply an emitting thermal structure that moves across the solar disk, but we are not interested in the actual identification of it with known waves in the solar atmosphere. There could be various possibilities for the propagating front such as EIT waves (Gallagher and Long , 2011), coronal waves related to chromospheric Moreton 145 waves (Narukage et al. , 2004) or others, which are known to perturb some structures like in the wave-filament interactions (Liu el al. , 2013). It is, however, not relevant to our studies to know which type of perturbation is seen.

Referee's comment:

line 35: why is flaring activity interesting ? section 2.1 this part is clearly written, but quite mathematical, and devoid of a connection with the physics. It would help to say the your spatio-temporal wakefield gets decomposed into a finite series of separable modes of time and space, which, in addition, are orthonormal, etc.

Authors' response:

Section 2.1 gives a short introduction to the basics of SVD; but we added the physical connection and the short description suggested by the referee.

Changes incorporated:

Page 3 Line 68:

In our case, SVD is used to decompose spatio-temporal data into a finite series of separable modes of time and space, which are orthonormal. The modes give the best representation of the relevant time and space scales of the data.

---

## Author Comment (AC2) · 27 Apr 2016

We thank the referee for the comments and we attach a PDF file with our detailed answers. The document provides a sequence of the referee's original comments (in blue), followed by our answer (in black) and the corresponding modification of the text (in red). We feel the modifications have enriched the paper with the discussions prompted by the comments. The transport analysis was extended to have evidence of diffusive transport. The new version will be uploaded as soon as the system requests us to do so, according to the journal procedures.

Please also note the supplement to this comment:
http://www.nonlin-processes-geophys-discuss.net/npg-2015-83/npg-2015-83-AC2-supplement.pdf

**Supplement:**

Referee #2

We thank the referee for his (her) valuable comments and suggestions. Following is our response and the changes incorporated in the new version of the manuscript.

Referee's comment

SUMMARY AND GENERAL EVALUATION
The authors present an interesting spatio-temporal analysis of 4-dim solar data, for a flaring event, the preflare phase, and a Quiet Sun region. They use a Singular Value Decomposition method and investigate the multi-scale behavior of the physical parameter of the temperature T, and attempt to extract information on the heat flow dT/dx in a flare. The description of the method is clear, and the data analysis steps are described in sufficient detail. In the second half of the paper the authors try to extract information on the heat flux before and during a flare, where they arrive at the result that diffusive transport is not relevant in solar flares, which is at odds with other studies. If the authors can reconcile the interpretation of this controversial result in terms of the limitations of the used method, the paper may possibly be suitable for publication.

Authors' response:

We do not really say that diffusion is not relevant in solar flares in general, we say that, for the particular event analyzed, transport is dominated by advection. However, we only considered transport along the main direction of propagation of the heat front ($x$). Prompted by the referee's comments, we extended the analysis to the propagation in the perpendicular direction ($y$), where advection is not expected to play an important role, in order to capture the diffusive contribution. As a result, we have found a correlation between the heat flux, $q_y$ and grad $T$ would allow, in principle, to estimate a diffusion coefficient. This additional results are included in the present version of the manuscript; we do not mention anymore that diffusion is not observed. Only for transport along $x$ it is subdominant.

Changes incorporated:

Page 18: The formulation for transport along $y$ was added with boldface font, starting with line 343 and Figure 15 was incorporated to illustrate the results.

Page 18 Line 332 added:

In contrast, the negative temperature gradient has no clear correlation with Q(x) which seems to indicate that diffusion is not the main drive for the heat flux along $x$ in this particular event.

DETAILED COMMENTS

Referee's comment:

In the Discussion and Conclusions the authors arrive at several important and some controversial results:
(1) "... a raw correlation between Fourier time scale and spatial scales during the flare, but not for pre-flare or quiet Sun. This may indicate that the flare-driven heat flows

tend to decay into smaller scales, in a cascade-like process." - Perhaps a reference and discussion can be made whether this is relevant to the scenario of "inverse MHD turbulent cascade" (e.g., Abramenko et al. 2003, ARep 47, 151, "Pre-Flare Changes in the Turbulence Regime for the Photospheric Magnetic Field in a Solar Active Region"; Antonucci et al. 1996, ApJ 456, 833 " Interpretation of the observed plasma turbulent velocities as a result of magnetic reconnection in solar flares"; LaRosa,T.N. and Moore,R.L. 1993, ApJ 418, 912-918, "A mechanism for bulk energization in the impulsive phase of solar flares: MHD turbulent cascade".

Authors' response:

We appreciate the referee's suggestion. It would certainly be interesting to do this discussion to enrich the paper. However, we have not included this because we decided to remove the interpretation of a cascading process, since, as pointed out by the other referee, this conclusion seems to be premature in the light of our results. We will take it into account for a future work on this subject.

Referee's comment:

(2) "... The pre-flare activity seems to produce larger low-amplitude fluctuations, characteristic of intermittency, which might herald the occurrence of the flare." — Discuss in this in the context of the paper by Abramenko et al. (2003), for instance.

Authors' response:

We thank the referee for the suggestion and we completely agree with its pertinence. We have included this discussion in the new version of the manuscript.

Changes incorporated:

Page 23 Line 412 added:
It is interesting to note that this result may be related to the findings of Abramenko et al. (2003) who also showed that there is evidence of intermittency in the magnetic field of an active region previous to the occurrence of a flare. They argue that this indicates that there is a turbulent phase before the flare, which would be in agreement with the intermittency in the temperature fluctuations found here.

Referee's comment:

(3) "A multi-scale analysis of the heat flux was also performed for the region associated with the flare. The thermal flux profiles along the main direction (x) of the flow were computed using original temperature maps and compared with the temperature variation along x, allowing to obtain the advection velocity profile. Diffusive transport is found to be not relevant." — This is contrary to other observational findings that the envelope of the flare area propagates like diffusive transport (Aschwanden 2012, ApJ 757, 94, "The spatio-temporal evolution of solar flares observed with AIA/SDO: Fractal diffusion, sub-diffusion, or logistic growth". The author's result may be biased because they use a 1-dim transport model (in x-direction), with no transport in transverse direction.

Authors' response:

We are actually not implying that there is no diffusive transport. What we find is that, *in this case*, transport is dominated by convection because of the strong heat pulse. However, as explained above, the analysis of transport along $y$ direction has modified this assertion. As pointed out by the referee, the 1D model produces a bias but this was offset by the addition of the transport in the perpendicular direction. Diffusive processes are much weaker than advection for transport along the direction of propagation of the heat front, but it is noticeable for transport in the perpendicular direction. We have commented this issue in the new version.

Changes incorporated:

Page 23 Line 420 added:

Diffusive transport is found to be sub-dominant and cannot be evaluated. A similar analysis was performed for transport along the direction perpendicular to the heat front propagation and in that case diffusion shows as an important contribution to transport.

Page 23 Line 428 added:

We point out that indications about a diffusive-like transport associated with a solar flare have been found by Aschwanden (2012) who actually found that the transport is sub-diffusive. This agrees with our result of Fig.7 which shows that the correlation of time and space scales corresponds with a sub-diffusive process.

Referee's comment:

Furthermore, they assume that there is no energy source inside the region of temperature maps (line 270). It would be more realistic to assume that the energy transport is isotropic in all directions, because the high-resolution images from AIA show that a flare entails many reconnection sites with magnetic field lines that go in all directions. Perhaps the authors can modify their heat flux model accordingly, or at least discuss the limitations of their model and what bias is expected in the diagnostics of the overall heat flux from a (point-like) flare source region.

Authors' response:

We are actually analyzing a small region very close to the brightest flaring region but that does not include it (see Fig. 3). The heat front moves away from the main flare site as it advances and penetrates into the region of study. The flare is not assumed to be point-like but rather it is outside our control volume. Therefore, it is valid to assume there are no energy sources inside the volume; most reconnection sites are in the flaring region which is outside. On the other hand, we cannot assume that transport is isotropic because of the strong contribution of advection which is directed along a specific direction in the case we study. But this issue is addressed by the modifications adding transport along $y$, as mentioned above. We have included a mention to the need of extending the model.

Changes incorporated:

Page 18 Line 343 added:

However, diffusive transport can be noticeable in the *y* direction where an important advection is not present. This can be studied with a similar analysis by averaging across the *x* direction. Notice that the averaging procedure is a simplification that would produce only approximate results.